# Maximum Entropy Inverse Reinforcement Learning of Diffusion Models with Energy-Based Models

**Sangwoong Yoon**[1], **Himchan Hwang**[2], **Dohyun Kwon**[1,3†], **Yung-Kyun Noh**[1,4†], **Frank C. Park**[2,5†]

[1]Korea Institute for Advanced Study, [2]Seoul National University, [3]University of Seoul,
[4]Hanyang University, [5]Saige Research, [†]Co-corresponding Authors
swyoon@kias.re.kr, himchan@robotics.snu.ac.kr,
dhkwon@uos.ac.kr, nohyung@hanyang.ac.kr, fcp@snu.ac.kr

## Abstract

We present a maximum entropy inverse reinforcement learning (IRL) approach for improving the sample quality of diffusion generative models, especially when the number of generation time steps is small. Similar to how IRL trains a policy based on the reward function learned from expert demonstrations, we train (or fine-tune) a diffusion model using the log probability density estimated from training data. Since we employ an energy-based model (EBM) to represent the log density, our approach boils down to the joint training of a diffusion model and an EBM. Our IRL formulation, named **Diffusion by Maximum Entropy IRL** (DxMI), is a minimax problem that reaches equilibrium when both models converge to the data distribution. The entropy maximization plays a key role in DxMI, facilitating the exploration of the diffusion model and ensuring the convergence of the EBM. We also propose **Diffusion by Dynamic Programming** (DxDP), a novel reinforcement learning algorithm for diffusion models, as a subroutine in DxMI. DxDP makes the diffusion model update in DxMI efficient by transforming the original problem into an optimal control formulation where value functions replace back-propagation in time. Our empirical studies show that diffusion models fine-tuned using DxMI can generate high-quality samples in as few as 4 and 10 steps. Additionally, DxMI enables the training of an EBM without MCMC, stabilizing EBM training dynamics and enhancing anomaly detection performance.

## 1   Introduction

Generative modeling is a form of imitation learning. Just as an imitation learner produces an action that mimics a demonstration from an expert, a generative model synthesizes a sample resembling the training data. In generative modeling, the expert to be imitated corresponds to the underlying data generation process. The intimate connection between generative modeling and imitation learning is already well appreciated in the literature [1, 2].

The connection to imitation learning plays a central role in diffusion models [3, 4], which generate samples by transforming a Gaussian noise through iterative additive refinements. The training of a diffusion model is essentially an instance of *behavioral cloning* [5], a widely adopted imitation learning algorithm that mimics an expert's action at each state. During training, a diffusion model is optimized to follow a predefined diffusion trajectory that interpolates between noise and data. The trajectory provides a step-by-step demonstration of how to transform Gaussian noise into a sample, allowing diffusion models to achieve a new state-of-the-art in many generation tasks.

Behavioral cloning is also responsible for the diffusion model's key limitation, the slow generation speed. A behavior-cloned policy is not reliable when the state distribution deviates from the expert demonstration [6, 7]. Likewise, the sample quality from a diffusion model degrades as the gap

38th Conference on Neural Information Processing Systems (NeurIPS 2024).

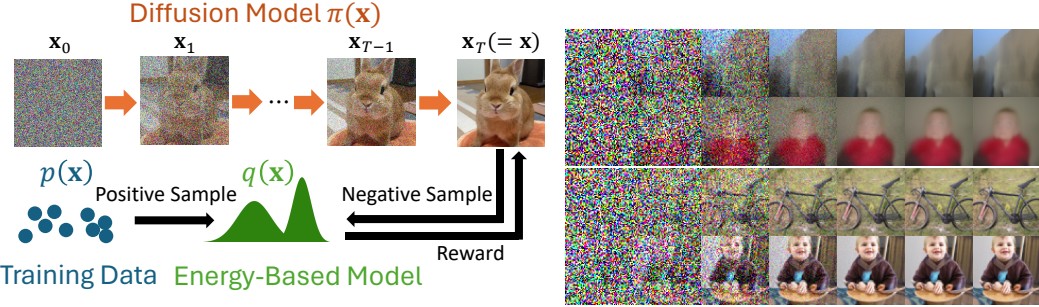

Figure 1: (Left) Overview of DxMI. The diffusion model $\pi(\mathbf{x})$ is trained using the energy of $q(\mathbf{x})$ as a reward. The EBM $q(\mathbf{x})$ is trained using samples from $\pi(\mathbf{x})$ as negative samples. (Right) ImageNet 64 generation examples from a 10-step diffusion model before DxMI fine-tuning (up) and after fine-tuning (down). Only the last six steps out of ten are shown.

between training and generation grows. A diffusion model is typically trained to follow a fine-grained diffusion trajectory of 1,000 or more steps. Since 1,000 neural network evaluations are prohibitively expensive, fewer steps are often used during generation, incurring the distribution shift from the training phase and thus degraded sample quality. Speeding up a diffusion model while maintaining its high sample quality is a problem of great practical value, becoming an active field of research [8, 9, 10, 11, 12, 13].

The slow generation in diffusion models can be addressed by employing inverse reinforcement learning (IRL; [14]), another imitation learning approach. Unlike behavioral cloning, which blindly mimics the expert's action at each state, the IRL approach first infers the reward function that explains the trajectory. When applied to diffusion models, IRL allows a faster generation trajectory to be found by guiding a sampler using the learned reward [11, 12, 13]. This approach is more frequently referred to as *adversarial* training because a common choice for the reward function is a discriminator classifying the training data and diffusion model's samples. However, resembling GAN, this adversarial approach may have similar drawbacks, such as limited exploration.

The first contribution of this paper is formulating *maximum entropy* IRL [15, 16, 2] for a diffusion model. Our formulation, named **Diffusion by Maximum Entropy IRL** (DxMI, pronounced "di-by-me"), is a minimax problem that jointly optimizes a diffusion model and an energy-based model (EBM). In DxMI, the EBM provides the estimated log density as the reward signal for the diffusion model. Then, the diffusion model is trained to maximize the reward from EBM while simultaneously maximizing the entropy of generated samples. The maximization of entropy in DxMI facilitates exploration and stabilizes the training dynamics, as shown in reinforcement learning (RL) [17], IRL [15], and EBM [18] literature. Furthermore, the entropy maximization lets the minimax problem have an equilibrium when both the diffusion model and the EBM become the data distribution.

The diffusion model update step in DxMI is equivalent to maximum entropy RL. However, this step is challenging to perform for two reasons. First, the diffusion model update requires the gradient of the marginal entropy of a diffusion model's samples, which is difficult to estimate for discrete-time diffusion models, such as DDPM [3]. Second, back-propagating the gradient through the whole diffusion model is often infeasible due to memory constraints. Even with sufficient memory, the gradient may explode or vanish during propagation through time, causing training instability [19].

Our second contribution is **Diffusion by Dynamic Programming** (DxDP), a novel maximum entropy RL algorithm for updating a diffusion model without the above-mentioned difficulties. First, DxDP mitigates the marginal entropy estimation issue by optimizing the upper bound of the original objective. In the upper bound, the marginal entropy is replaced by conditional entropies, which are easier to compute for a diffusion model. Then, DxDP removes the need for back-propagation in time by interpreting the objective as an optimal control problem and applying dynamic programming using value functions. The connection between optimal control and diffusion models is increasingly gaining attention [20, 21, 22], but DxDP is the first instance of applying discrete-time dynamic programming directly to train a diffusion model. Compared to policy gradient methods, we empirically find that DxDP converges faster and provides stronger diffusion models. As an RL algorithm, DxDP may have broader utility, such as fine-tuning a diffusion model from human or AI feedback.

We provide experimental results demonstrating the effectiveness of DxMI in training diffusion models and EBMs. On image generation tasks, DxMI can train strong short-run diffusion models that generate samples in 4 or 10 neural network evaluations. Also, DxMI can be used to train strong energy-based anomaly detectors. Notably, DxMI provides a novel way to train EBM without MCMC, which is computationally demanding and sensitive to the choice of hyperparameters.

The paper is structured as follows. In Section 2, we introduce the necessary preliminaries. Section 3 presents the DxMI framework, and Section 4 proposes the DxDP algorithm. Experimental results and related work are provided in Sections 5 and 6, respectively. Section 7 concludes the paper. The code for DxMI can be found in `https://github.com/swyoon/Diffusion-by-MaxEntIRL.git`.

## 2 Preliminaries

**Diffusion Models.** The diffusion model refers to a range of generative models trained by reversing the trajectory from the data distribution to the noise distribution. Among diffusion models, we focus on discrete-time stochastic samplers, such as DDPM [3], which synthesize a sample through the following iteration producing $\mathbf{x}_0, \mathbf{x}_1, \ldots, \mathbf{x}_T \in \mathbb{R}^D$:

$$\mathbf{x}_0 \sim \mathcal{N}(0, I) \quad \text{and} \quad \mathbf{x}_{t+1} = a_t \mathbf{x}_t + \mu(\mathbf{x}_t, t) + \sigma_t \epsilon_t \quad \text{for} \quad t = 0, 1, \ldots, T-1, \tag{1}$$

where $\epsilon_t \sim \mathcal{N}(0, I)$ and $\mu(\mathbf{x}, t)$ is the output of a neural network. The coefficients $a_t \in \mathbb{R}$ and $\sigma_t \in \mathbb{R}_{>0}$ are constants. Note that we reverse the time direction in a diffusion model from the convention to be consistent with RL. The final state $\mathbf{x}_T$ is the sample generated by the diffusion model, and its marginal distribution is $\pi(\mathbf{x}_T)$. We will often drop the subscript $T$ and write the distribution as $\pi(\mathbf{x})$. The conditional distribution of a transition in Eq. (1) is denoted as $\pi(\mathbf{x}_{t+1}|\mathbf{x}_t)$. For continuous-time diffusion models [23], Eq. (1) corresponds to using the Euler-Maruyama solver.

The generation process in Eq. (1) defines a $T$-horizon Markov Decision Process (MDP) except for the reward. State $\mathbf{s}_t$ and action $\mathbf{a}_t$ are defined as $\mathbf{s}_t = (\mathbf{x}_t, t)$ and $\mathbf{a}_t = \mathbf{x}_{t+1}$. The transition dynamics is defined as $p(\mathbf{s}_{t+1}|\mathbf{s}_t, \mathbf{a}_t) = \delta_{(\mathbf{a}_t, t+1)}$, where $\delta_{(\mathbf{x}_t, t)}$ is a Dirac delta function at $(\mathbf{x}_t, t)$. With a reward function defined, a diffusion model can be trained with RL [11, 24, 19, 25]. In this paper, we consider a case where the reward is the log data density $\log p(\mathbf{x})$, which is unknown.

**Energy-Based Models.** An energy-based model (EBM) $q(\mathbf{x})$ uses a scalar function called an energy $E(\mathbf{x})$ to represent a probability distribution:

$$q(\mathbf{x}) = \frac{1}{Z} \exp(-E(\mathbf{x})/\tau), \quad E : \mathcal{X} \to \mathbb{R}, \tag{2}$$

where $\tau > 0$ is temperature, $\mathcal{X}$ is the compact domain of data, and $Z = \int_{\mathcal{X}} \exp(-E(\mathbf{x})/\tau)d\mathbf{x} < \infty$ is the normalization constant.

The standard method for training an EBM is by minimizing KL divergence between data $p(\mathbf{x})$ and EBM $q(\mathbf{x})$, i.e., $\min_q KL(p||q)$, where $KL(p||q) := \int_{\mathcal{X}} p(\mathbf{x}) \log(p(\mathbf{x})/q(\mathbf{x}))d\mathbf{x}$. Computing the gradient of $KL(p||q)$ with respect to EBM parameters requires MCMC sampling, which is computationally demanding and sensitive to hyperparameters. The algorithm presented in this paper serves as an alternative method for training an EBM without MCMC.

EBMs have a profound connection to maximum entropy IRL, where Eq. (2) serves as a model for an expert's policy [15, 16, 2]. In maximum entropy IRL, $\mathbf{x}$ corresponds to an action (or a sequence of actions), and $E(\mathbf{x})$ represents the expert's cost of the action. The expert is then assumed to generate actions following $q(\mathbf{x})$. This assumption embodies the maximum entropy principle because Eq. (2) is a distribution that minimizes the cost while maximizing the entropy of the action. Here, $\tau$ balances cost minimization and entropy maximization.

## 3 Diffusion by Maximum Entropy Inverse Reinforcement Learning

### 3.1 Objective: Generalized Contrastive Divergence

We aim to minimize the (reverse) KL divergence between a diffusion model $\pi(\mathbf{x})$ and the data density $p(\mathbf{x})$. This minimization can improve the sample quality of $\pi(\mathbf{x})$, particularly when $T$ is small.

$$\min_{\pi \in \Pi} KL(\pi(\mathbf{x})||p(\mathbf{x})) = \max_{\pi \in \Pi} \mathbb{E}_\pi[\log p(\mathbf{x})] + \mathcal{H}(\pi(\mathbf{x})), \tag{3}$$

where $\Pi$ is the set of feasible $\pi(\mathbf{x})$'s, and $\mathcal{H}(\pi) = -\int \pi(\mathbf{x}) \log \pi(\mathbf{x}) d\mathbf{x}$ is the differential entropy. This minimization is a maximum entropy RL problem: The log data density $\log p(\mathbf{x})$ is the reward, and $\pi(\mathbf{x})$ is the stochastic policy. However, we cannot solve Eq. (3) directly since $\log p(\mathbf{x})$ is unknown in a typical generative modeling setting. Instead, training data from $p(\mathbf{x})$ are available, allowing us to employ an *Inverse* RL approach.

In this paper, we present **Diffusion by Maximum Entropy IRL (DxMI)** as an IRL approach for solving Eq. (3). We employ an EBM $q(\mathbf{x})$ (Eq. (2)) as a surrogate for $p(\mathbf{x})$ and use $\log q(\mathbf{x})$ as the reward for training the diffusion model instead of $\log p(\mathbf{x})$. At the same time, we train $q(\mathbf{x})$ to be close to $p(\mathbf{x})$ by minimizing the divergence between $p(\mathbf{x})$ and $q(\mathbf{x})$:

$$\min_{\pi \in \Pi} KL(\pi(\mathbf{x})||q(\mathbf{x})) \quad \text{and} \quad \min_{q \in \mathcal{Q}} KL(p(\mathbf{x})||q(\mathbf{x})), \tag{4}$$

where $\mathcal{Q}$ is the feasible set of EBMs. When the two KL divergences become zero, $p(\mathbf{x}) = q(\mathbf{x}) = \pi(\mathbf{x})$ is achieved. However, $\min_{q \in \mathcal{Q}} KL(p(\mathbf{x})||q(\mathbf{x}))$ is difficult due to the normalization constant of $q(\mathbf{x})$. Instead, we consider an alternative minimax formulation inspired by Contrastive Divergence (CD; [26]), a celebrated algorithm for training an EBM.

---

**Objective.** Let $p(\mathbf{x})$ be the data distribution. Suppose that $\mathcal{Q}$ and $\Pi$ are the feasible sets of the EBM $q(\mathbf{x})$ and the diffusion model $\pi(\mathbf{x})$, respectively. The learning problem of DxMI is formulated as follows:

$$\min_{q \in \mathcal{Q}} \max_{\pi \in \Pi} KL(p(\mathbf{x})||q(\mathbf{x})) - KL(\pi(\mathbf{x})||q(\mathbf{x})). \tag{5}$$

---

We shall call Eq. (5) *Generalized* CD (GCD), because Eq. (5) generalizes CD by incorporating a general class of samplers. CD [26] is originally given as $\min_{q \in \mathcal{Q}} KL(p(\mathbf{x})||q(\mathbf{x})) - KL(\nu_{p,q}^k(\mathbf{x})||q(\mathbf{x}))$. Here, $\nu_{p,q}^k(\mathbf{x})$ is a $k$-step MCMC sample distribution where Markov chains having $q(\mathbf{x})$ as a stationary distribution are initialized from $p(\mathbf{x})$. GCD replaces $\nu_{p,q}^k(\mathbf{x})$ with a general sampler $\pi(\mathbf{x})$ at the expense of introducing a max operator.

When the models are well-specified, i.e., $p(\mathbf{x}) \in \mathcal{Q} = \Pi$, Nash equilibrium of GCD is $p(\mathbf{x}) = q(\mathbf{x}) = \pi(\mathbf{x})$, which is identical to the solution of Eq. (4). Meanwhile, there is no need to compute the normalization constant, as the two KL divergences cancel the normalization constant out. Note that the objective function (Eq. (5)) can be negative, allowing $q(\mathbf{x})$ be closer to $p(\mathbf{x})$ than $\pi(\mathbf{x})$.

Our main contribution is exploring the application of discrete-time diffusion models (Eq. (1)) as $\pi(\mathbf{x})$ in GCD and not discovering GCD for the first time. GCD is mathematically equivalent to a formulation called variational maximum likelihood or adversarial EBM, which has appeared several times in EBM literature [27, 28, 29, 30, 31, 32, 33]. However, none of them have investigated the use of a diffusion model as $\pi(\mathbf{x})$, where optimization and entropy estimation are challenging. We discuss the challenges in Section 3.2 and provide a novel algorithm to address them in Section 4.

### 3.2 Alternating Update of EBM and Diffusion Model

In DxMI, we update a diffusion model and an EBM in an alternative manner to find the Nash equilibrium. We write $\theta$ and $\phi$ as the parameters of the energy $E_\theta(\mathbf{x})$ and a diffusion model $\pi_\phi(\mathbf{x})$, respectively. While EBM update is straightforward, we require a subroutine described in Section 4 for updating the diffusion model. The entire procedure of DxMI is summarized in Algorithm 1.

**EBM Update.** The optimization with respect to EBM is written as $\min_\theta \mathbb{E}_{p(\mathbf{x})}[E_\theta(\mathbf{x})] - \mathbb{E}_{\pi_\phi(\mathbf{x})}[E_\theta(\mathbf{x})]$. During the update, we also regularize the energy by the square of energies $\gamma(\mathbb{E}_{p(\mathbf{x})}[E_\theta(\mathbf{x})^2] + \mathbb{E}_{\pi_\phi(\mathbf{x})}[E_\theta(\mathbf{x})^2])$ for $\gamma > 0$ to ensure the energy is bounded. We set $\gamma = 1$ unless stated otherwise. This regularizer is widely adopted in EBM [34, 35].

**Difficulty of Diffusion Model Update.** Ideally, diffusion model parameter $\phi$ should be updated by minimizing $KL(\pi_\phi||q_\theta) = \mathbb{E}_{\pi_\phi(\mathbf{x})}[E_\theta(\mathbf{x})/\tau] - \mathcal{H}(\pi_\phi(\mathbf{x}))$. However, this update is difficult in practice for two reasons.

First, marginal entropy $\mathcal{H}(\pi_\phi(\mathbf{x}))$ is difficult to estimate. Discrete-time diffusion models (Eq. (1)) do not provide an efficient way to evaluate $\log \pi_\phi(\mathbf{x})$ required in the computation of $\mathcal{H}(\pi_\phi(\mathbf{x}))$, unlike some continuous-time models, e.g., continuous normalizing flows [36, 23]. Other entropy estimators based on $k$-nearest neighbors [37] or variational methods [38, 39] do not scale well to

---

**Algorithm 1** Diffusion by Maximum Entropy IRL

---

1: **Input:** Dataset $\mathcal{D}$, Energy $E_\theta(\mathbf{x})$, Value $V_\psi^t(\mathbf{x}_t)$, and Sampler $\pi_\phi(\mathbf{x}_{0:T})$
2: $s_t \leftarrow \sigma_t$            // Initialize Adaptive Velocity Regularization (AVR)
3: **for** $\mathbf{x}$ in $\mathcal{D}$ **do**           // Minibatch dimension is omitted for brevity.
4:     Sample $\mathbf{x}_{0:T} \sim \pi_\phi(\mathbf{x}_{0:T})$.
5:     $\min_\theta E_\theta(\mathbf{x}) - E_\theta(\mathbf{x}_T) + \gamma(E_\theta(\mathbf{x})^2 + E_\theta(\mathbf{x}_T)^2)$       // Energy update
6:     **for** $t = T-1, \ldots, 0$ **do**           // Value update
7:         $\min_\psi \left[ \text{sg}[V_\psi^{t+1}(\mathbf{x}_{t+1})] + \tau \log \pi(\mathbf{x}_{t+1}|\mathbf{x}_t) + \frac{\tau}{2s_t^2}||\mathbf{x}_t - \mathbf{x}_{t+1}||^2 - V_\psi^t(\mathbf{x}_t) \right]^2$
8:     **end for**
9:     **for** $\mathbf{x}_t$ randomly chosen among $\mathbf{x}_{0:T}$ **do**         // Sampler update
10:         Sample one-step: $\mathbf{x}_{t+1} \sim \pi_\phi(\mathbf{x}_{t+1}|\mathbf{x}_t)$       // Reparametrization trick
11:         $\min_\phi V_\psi^{t+1}(\mathbf{x}_{t+1}(\phi)) + \tau \log \pi(\mathbf{x}_{t+1}|\mathbf{x}_t) + \frac{\tau}{2s_t^2}||\mathbf{x}_t - \mathbf{x}_{t+1}(\phi)||^2$ // $\mathbf{x}_{t+1}$ is a function of $\phi$.
12:     **end for**
13:     $s_t^2 \leftarrow \alpha s_t^2 + (1-\alpha)||\mathbf{x}_t - \mathbf{x}_{t+1}||^2/D$         // AVR update
14: **end for**

---

high-dimensional spaces. Second, propagating the gradient through time in a diffusion model may require significant memory. Also, the gradient may explode or vanish during propagation, making the training unstable [19].

## 4 Diffusion by Dynamic Programming

In this section, we present a novel RL algorithm for a diffusion model, **Diffusion by Dynamic Programming** (DxDP), which addresses the difficulties in updating a diffusion model for the reward. DxDP leverages optimal control formulation and value functions to perform the diffusion model update in DxMI efficiently. Note that DxDP can be used separately from DxMI to train a diffusion model for an arbitrary reward.

### 4.1 Optimal Control Formulation

Instead of solving $\min_\phi KL(\pi_\phi(\mathbf{x}_T)||q_\theta(\mathbf{x}_T))$ directly, we minimize its upper bound obtained from the data processing inequality:

$$KL(\pi_\phi(\mathbf{x}_T)||q_\theta(\mathbf{x}_T)) \leq KL(\pi_\phi(\mathbf{x}_{0:T})||q_\theta(\mathbf{x}_T)\tilde{q}(\mathbf{x}_{0:T-1}|\mathbf{x}_T)). \quad (6)$$

Here, we introduce an auxiliary distribution $\tilde{q}(\mathbf{x}_{0:T-1}|\mathbf{x}_T)$, and the inequality holds for an arbitrary choice of $\tilde{q}(\mathbf{x}_{0:T-1}|\mathbf{x}_T)$. In this paper, we consider a particularly simple case where $\tilde{q}(\mathbf{x}_{0:T-1}|\mathbf{x}_T)$ is factorized into conditional Gaussians as follows:

$$\tilde{q}(\mathbf{x}_{0:T-1}|\mathbf{x}_T) = \prod_{t=0}^{T-1} \tilde{q}(\mathbf{x}_t|\mathbf{x}_{t+1}), \text{ where } \tilde{q}(\mathbf{x}_t|\mathbf{x}_{t+1}) = \mathcal{N}(\mathbf{x}_{t+1}, s_t^2 I), \quad s_t > 0. \quad (7)$$

Now we minimize the right-hand side of Eq. (6): $\min_\phi KL(\pi_\phi(\mathbf{x}_{0:T})||q_\theta(\mathbf{x}_T)\tilde{q}(\mathbf{x}_{0:T-1}|\mathbf{x}_T))$. When we plug in the definitions of each distribution, multiply by $\tau$, and discard all constants, we obtain the following problem:

$$\min_\phi \mathbb{E}_{\pi_\phi(\mathbf{x}_{0:T})} \left[ E_\theta(\mathbf{x}_T) + \tau \sum_{t=0}^{T-1} \log \pi_\phi(\mathbf{x}_{t+1}|\mathbf{x}_t) + \tau \sum_{t=0}^{T-1} \frac{1}{2s_t^2}||\mathbf{x}_{t+1} - \mathbf{x}_t||^2 \right], \quad (8)$$

which is an optimal control problem. The controller $\pi_\phi(\cdot)$ is optimized to minimize the terminal cost $E_\theta(\mathbf{x}_T)$ plus the running costs for each transition $(\mathbf{x}_t, \mathbf{x}_{t+1})$. The first running cost $\log \pi_\phi(\mathbf{x}_{t+1}|\mathbf{x}_t)$ is responsible for conditional entropy maximization, because $\mathbb{E}_\pi[\log \pi_\phi(\mathbf{x}_{t+1}|\mathbf{x}_t)] = -\mathcal{H}(\pi_\phi(\mathbf{x}_{t+1}|\mathbf{x}_t))$. The second running cost regularizes the "velocity" $||\mathbf{x}_{t+1} - \mathbf{x}_t||^2$. The temperature $\tau$ balances between the terminal and running costs.

We have circumvented the marginal entropy computation in GCD, as all terms in Eq. (8) are easily computable. For the diffusion model considered in this paper (Eq. (1)), the conditional entropy has a particularly simple expression $\mathcal{H}(\pi_\phi(\mathbf{x}_{t+1}|\mathbf{x}_t)) = D \log \sigma_t + 0.5D \log 2\pi$. Therefore, optimizing the entropy running cost amounts to learning $\sigma_t$'s in diffusion, and we treat $\sigma_t$'s as a part of the diffusion model parameter $\phi$ in DxMI.

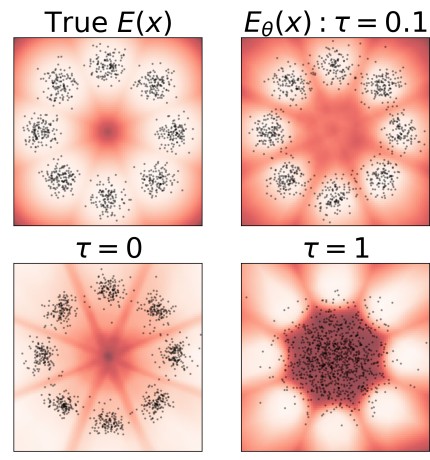

True $E(x)$     $E_\theta(x) : \tau = 0.1$

$\tau = 0$     $\tau = 1$

Figure 2: 2D density estimation on 8 Gaussians. Red shades indicate the energy (white is low), and the dots are generated samples.

Table 1: Quantitative results for 8 Gaussians experiment. $SW$ denotes the sliced Wasserstein distance between samples and data. AUC is computed for classification between data and uniform noise using the energy. The standard deviation is computed from 5 independent samplings. The ideal maximum value of AUC is about 0.906.

| Method | T | Pretrain | $\tau$ | $SW$ ($\downarrow$) | AUC ($\uparrow$) |
|---|---|---|---|---|---|
| DDPM | 5 | - | - | 0.967±0.005 | - |
| DDPM | 10 | - | - | 0.824±0.002 | - |
| DDPM | 100 | - | - | 0.241±0.003 | - |
| DDPM | 1000 | - | - | 0.123±0.014 | - |
| DxMI | 5 | ○ | 0 | 0.074±0.018 | 0.707 |
| DxMI | 5 | ○ | 0.01 | 0.074±0.017 | 0.751 |
| DxMI | 5 | ○ | 0.1 | **0.068±0.004** | **0.898** |
| DxMI | 5 | ○ | 1 | 1.030±0.004 | 0.842 |
| DxMI | 5 | × | 0.1 | 0.076±0.011 | 0.883 |

## 4.2 Dynamic Programming

We propose a dynamic programming approach for solving Eq. (8). Dynamic programming introduces value functions to break down the problem into smaller problems at each timestep, removing the need for back-propagation in time. Then, a policy, a diffusion model in our case, is optimized through iterative alternating applications of policy evaluation and policy improvement steps.

**Value Function.** A value function, or cost-to-go function $V_\psi^t(\mathbf{x}_t)$, is defined as the expected sum of the future costs starting from $\mathbf{x}_t$, following $\pi$. We write the parameters of a value function as $\psi$.

$$V_\psi^t(\mathbf{x}_t) = \mathbb{E}_\pi \left[ E_\theta(\mathbf{x}_T) + \tau \sum_{t'=t}^{T-1} \log \pi_\phi(\mathbf{x}_{t'+1}|\mathbf{x}_{t'}) + \sum_{t'=t}^{T-1} \frac{\tau}{2s_{t'}^2} ||\mathbf{x}_{t'+1} - \mathbf{x}_{t'}||^2 \middle| \mathbf{x}_t \right], \quad (9)$$

for $t = 0, \ldots, T-1$. Note that $V^T(\mathbf{x}_T) = E(\mathbf{x}_T)$. A value function can be implemented with a neural network, but there are multiple design choices, such as whether to share the parameters with $\pi(\mathbf{x})$ or $E(\mathbf{x})$, and also whether the parameters should be shared across time. We explore the options in our experiments.

**Policy Evaluation.** During policy evaluation, we estimate the value function for the current diffusion model by minimizing the Bellman residual, resulting in the temporal difference update.

$$\min_\psi \mathbb{E}_{\mathbf{x}_t, \mathbf{x}_{t+1} \sim \pi}[(\text{sg}[V_\psi^{t+1}(\mathbf{x}_{t+1})] + \tau \log \pi_\phi(\mathbf{x}_{t+1}|\mathbf{x}_t) + \frac{\tau}{2s_t^2} ||\mathbf{x}_t - \mathbf{x}_{t+1}||^2 - V_\psi^t(\mathbf{x}_t))^2], \quad (10)$$

where $\text{sg}[\cdot]$ denotes a stop-gradient operator indicating that gradient is not computed for the term.

**Policy Improvement.** The estimated value is used to improve the diffusion model. For each $\mathbf{x}_t$ in a trajectory $\mathbf{x}_{0:T}$ sampled from $\pi_\phi(\mathbf{x}_{0:T})$, the diffusion model is optimized to minimize the next-state value and the running costs.

$$\min_\phi \mathbb{E}_{\pi_\phi(\mathbf{x}_{t+1}|\mathbf{x}_t)} \left[ V_\psi^{t+1}(\mathbf{x}_{t+1}) + \tau \log \pi_\phi(\mathbf{x}_{t+1}|\mathbf{x}_t) + \frac{\tau}{2s_t^2} ||\mathbf{x}_t - \mathbf{x}_{t+1}||^2 \middle| \mathbf{x}_t \right]. \quad (11)$$

In practice, each iteration of policy evaluation and improvement involves a single gradient step.

**Adaptive Velocity Regularization (AVR).** We additionally propose a method for systematically determining the hyperparameter $s_t$'s of the auxiliary distribution $\tilde{q}(\mathbf{x}_{0:T-1}|\mathbf{x}_T)$. We can optimize $s_t$ such that the inequality Eq. (6) is as tight as possible by solving $\min_{s_0,\ldots,s_{T-1}} KL(\pi_\phi(\mathbf{x}_{0:T})||q_\theta(\mathbf{x}_T)\tilde{q}(\mathbf{x}_{0:T-1}|\mathbf{x}_T))$. After calculation (details in Appendix A), the optimal $s_t^*$ can be obtained analytically: $(s_t^*)^2 = \mathbb{E}_{\mathbf{x}_t, \mathbf{x}_{t+1} \sim \pi}[||\mathbf{x}_t - \mathbf{x}_{t+1}||^2]/D$. In practice, we can use exponential moving average to compute the expectation $\mathbb{E}_{\mathbf{x}_t, \mathbf{x}_{t+1} \sim \pi}[||\mathbf{x}_t - \mathbf{x}_{t+1}||^2]$ during training: $s_t^2 \leftarrow \alpha s_t^2 + (1-\alpha)||\mathbf{x}_t - \mathbf{x}_{t+1}||^2/D$ where we set $\alpha = 0.99$ for all experiment.

### 4.3 Techniques for Image Generation Experiments

When using DxDP for image generation, one of the most common applications of diffusion models, we introduce several design choices to DxDP to enhance performance and training stability. The resulting algorithm is summarized in Algorithm 2.

**Time-Independent Value Function.** In image generation experiments (Section 5.2), we let the value function be independent of time, i.e., $V_\psi^t(\mathbf{x}_t) = V_\psi(\mathbf{x}_t)$. Removing the time dependence reduces the number of parameters to be trained. More importantly, a time-independent value function can learn better representation because the value function is exposed to diverse inputs, including both noisy and clean images. On the contrary, a time-dependent value function $V_\psi^t(\mathbf{x}_t)$ never observes samples having different noise levels than the noise level of $\mathbf{x}_t$.

**Time Cost.** Also, in the value update (Eq. (10)) step of image generation experiments, we introduce *time cost* function $R(t) > 0$, which replaces the running cost terms $\tau \log \pi_\phi(\mathbf{x}_{t+1}|\mathbf{x}_t) + \tau||\mathbf{x}_t - \mathbf{x}_{t+1}||^2/(2s_t^2)$. The time cost $R(t)$ only depends on time $t$. The modified value update equation is given as follows:

$$\min_\psi \mathbb{E}_{\mathbf{x}_t, \mathbf{x}_{t+1} \sim \pi}[(\text{sg}[V_\psi(\mathbf{x}_{t+1})] + R(t) - V_\psi(\mathbf{x}_t))^2]. \tag{12}$$

Meanwhile, we retain the running cost terms in the diffusion model (policy) update step (Eq. (11)). The time cost $R(t)$ is predetermined and fixed throughout training. The introduction of time cost is motivated by the observation that the running costs can fluctuate during the initial stage of training, posing difficulty in value function learning. The time cost stabilizes the training by reducing this variability. Moreover, the time cost ensures that the value function decreases monotonically over time. Such monotonicity is known to be beneficial in IRL for episodic tasks [16].

We employ two types of $R(t)$: "linear" and "sigmoid". A *linear* time cost is given as $R(t) = c$ where we use $c = 0.05$. The linear time cost encourages the value to decrease linearly as time progresses. The *sigmoid* time cost is $R(t) = \sigma(-t+T/2)) - \sigma(-t-1+T/2))$, where $\sigma(x) = (1+\exp(-x))^{-1}$. With the sigmoid time cost, the value function is trained to follow a sigmoid function centered at $T/2$ when plotted against the time. Other forms of $R(t)$ are also possible.

**Separate tuning of $\tau$.** In image generation, we assign different values of temperature $\tau$ for entropy regularization $\log \pi_\phi(\mathbf{x}_{t+1}|\mathbf{x}_t)$ and velocity regularization $||\mathbf{x}_t - \mathbf{x}_{t+1}||^2/(2s_t^2)$, such that the resulting running cost becomes $\tau_1 \log \pi_\phi(\mathbf{x}_{t+1}|\mathbf{x}_t) + \tau_2||\mathbf{x}_t - \mathbf{x}_{t+1}||^2/(2s_t^2)$. Typically, we found $\tau_1 > \tau_2$ beneficial, indicating the benefit of exploration. Setting $\tau_1 \neq \tau_2$ does not violate our maximum entropy formulation, as scaling $\tau_2$ is equivalent to scaling $s_t^2$'s, which can be set arbitrarily.

## 5 Experiments

In this section, we provide empirical studies that demonstrate the effectiveness of DxMI in training a diffusion model and an EBM. We first present a 2D example, followed by image generation and anomaly detection experiments. More details on experiments can be found in Appendix C.

### 5.1 2D Synthetic Data

We illustrate how DxMI works on 2D 8 Gaussians data. DxMI is applied to train a five-step diffusion model ($T = 5$) with a corresponding time-dependent value network, both parametrized by time-conditioned multi-layer perceptron (MLP). The last time step ($T = 5$) of the value network is treated as the energy. The sample quality is measured with sliced Wasserstein distance (SW) to test data. Also, we quantify the quality of an energy function through the classification performance to uniform noise samples (Table 1).

First, we investigate the effect of maximum entropy regularization $\tau$. Setting an appropriate value for $\tau$ greatly benefits the quality of both the energy and the samples. When $\tau = 0.1$, the samples from DxMI have smaller SW than the samples from a full-length DDPM do. The energy also accurately captures the data distribution, scoring high AUC against the uniform noise. Without entropy regularization ($\tau = 0$), DxMI becomes similar to GAN [40]. The generated samples align moderately well with the training data, but the energy does not reflect the data distribution. When $\tau$ is too large ($\tau = 1$), the generated samples are close to noise. In this regime, DxMI behaves similarly

Table 2: CIFAR-10 unconditional image generation. †: the starting point of DxMI fine-tuning.

| | NFE | FID ($\downarrow$) | Rec. ($\uparrow$) |
|---|---|---|---|
| Score SDE (VE) [23] | 2000 | 2.20 | 0.59 |
| PD [10] | 8 | 2.57 | - |
| Consistency Model [42] | 2 | 2.93 | - |
| PD [10] | 1 | 8.34 | - |
| 2-Rectified Flow [43] | 1 | 4.85 | 0.50 |
| Consistency Model [42] | 1 | 3.55 | - |
| StyleGAN-XL [44] | 1 | 1.85 | 0.47 |
| **Backbone: DDPM** | | | |
| DDPM [3] | 1000 | 3.21 | 0.57 |
| FastDPM† [8] | 10 | 35.85 | 0.29 |
| DDIM [45] | 10 | 13.36 | - |
| SFT-PG [11] | 10 | 4.82 | 0.606 |
| DxMI | 10 | 3.19 | 0.625 |
| $\quad \tau = 0$ | 10 | 3.77 | 0.613 |
| $\quad$ Linear time cost | 10 | 3.39 | 0.595 |
| $\quad$ No time cost | 10 | 5.18 | 0.595 |
| DxMI + Value Guidance | 10 | 3.17 | 0.623 |
| $\quad \tau = 0$ | 10 | 3.72 | 0.613 |
| **Backbone: DDGAN** | | | |
| DDGAN† [46] | 4 | 4.15 | 0.523 |
| DxMI | 4 | 3.65 | 0.532 |

Table 3: ImageNet $64\times64$ conditional image generation. †: the starting point of DxMI fine-tuning.

| | NFE | FID ($\downarrow$) | Prec. ($\uparrow$) | Rec. ($\uparrow$) |
|---|---|---|---|---|
| ADM [47] | 250 | 2.07 | 0.74 | 0.63 |
| DFNO [48] | 1 | 8.35 | - | - |
| PD [10] | 1 | 15.39 | 0.59 | 0.62 |
| BigGAN-deep [49] | 1 | 4.06 | 0.79 | 0.48 |
| **Backbone: EDM** | | | | |
| EDM (Heun) [50] | 79 | 2.44 | 0.71 | 0.67 |
| EDM (Ancestral)† | 10 | 50.27 | 0.37 | 0.35 |
| EDM (Ancestral)† | 4 | 82.95 | 0.26 | 0.25 |
| Consistency Model [42] | 2 | 4.70 | 0.69 | 0.64 |
| Consistency Model [42] | 1 | 6.20 | 0.68 | 0.63 |
| DxMI | 10 | 2.68 | 0.777 | 0.574 |
| $\quad \tau = 0$ | 10 | 2.72 | 0.782 | 0.564 |
| $\quad$ Linear time cost | 10 | 2.81 | 0.742 | 0.594 |
| DxMI+Value Guidance | 10 | 2.67 | 0.780 | 0.574 |
| $\quad \tau = 0$ | 10 | 2.76 | 0.786 | 0.560 |
| DxMI | 4 | 3.21 | 0.758 | 0.568 |
| $\quad \tau = 0$ | 4 | 3.65 | 0.767 | 0.552 |
| $\quad$ Linear time cost | 4 | 3.40 | 0.762 | 0.554 |
| DxMI+Value Guidance | 4 | 3.18 | 0.763 | 0.566 |
| $\quad \tau = 0$ | 4 | 3.67 | 0.770 | 0.541 |

to Noise Contrastive Estimation [41], enabling energy function learning to a certain extent. These effects are visualized in Fig. 2.

Next, we experiment on whether pre-training a sampler as DDPM helps DxMI. Table 1 suggests that the pre-training is beneficial but not necessary to make DxMI work. We also visualize the value functions in Fig. 3 and find that the time evolution of value interpolates the data distribution and a Gaussian distribution.

## 5.2 Image Generation: Training Diffusion Models with Small $T$

On image generation tasks, we show that DxMI can be used to fine-tune a diffusion model with reduced generation steps, such as $T = 4$ or 10. We test DxMI on unconditional CIFAR-10 [52] ($32 \times 32$), conditional ImageNet [53] downsampled to $64 \times 64$, and LSUN Bedroom [54] ($256\times256$), using three diffusion model backbones, DDPM [3], DDGAN [46], and variance exploding version of EDM [50]. The results can be found in Table 2, 3, and 4. Starting from

Table 4: LSUN Bedroom $256 \times 256$ unconditional image generation.

| | NFE | FID ($\downarrow$) | Prec. ($\uparrow$) | Rec. ($\uparrow$) |
|---|---|---|---|---|
| StyleGAN2 [51] | 1 | 2.35 | 0.59 | 0.48 |
| **Backbone: EDM** | | | | |
| EDM [50] | 79 | 2.44 | 0.71 | 0.67 |
| Consistency Model [42] | 2 | 5.22 | 0.68 | 0.39 |
| DxMI | 4 | 5.93 | 0.563 | 0.477 |

a publicly available checkpoint of each pretrained backbone, we first adjust the noise schedule for the target sampling steps $T$. When adjusting the noise, for DDPM, we follow the schedule of FastDPM [8], and for EDM, we use Eq. (5) of [50]. No adjustment is made for DDGAN, which was originally built for $T = 4$. The adjusted models are used as the starting point of DxMI training. A single CIFAR-10 run reaches the best FID in less than 4 hours on four A100 GPUs. We set $\tau_1 = 0.1$ and $\tau_2 = 0.01$. The sigmoid time cost is used for all image generation experiments. The sample quality is measured by FID [55], Precision (Prec., [56]), and Recall (Rec., [56]). ResNet is used as our value function and is trained from scratch. More experimental details are in Appendix C.2.

Short-run diffusion models fine-tuned by DxMI display competitive sample quality. Unlike distillation methods, which are often limited by their teacher model's performance, DxMI can surpass the pre-trained starting point. Although DxMI does not support single-step generation, DxMI offers a principled approach to training a high-quality generative model with a moderate computation burden

(Appendix D). Note that DDGAN does not fit the formulation of DxMI, as $\pi(\mathbf{x}_{t+1}|\mathbf{x}_t)$ in DDGAN is not Gaussian. Nevertheless, DxMI can still enhance sample quality, showing its robustness.

Furthermore, DxMI outperforms SFT-PG [11], another IRL approach implemented with a policy gradient. For a fair comparison, we have ensured that the backbone and the initial checkpoint of SFT-PG and DxMI are identical. Thus, the performance gap can be attributed to the two differences between SFT-PG and DxMI. First, DxMI uses dynamic programming instead of policy gradient. In DxDP, the value function is more directly utilized to guide the learning of the diffusion model. Meanwhile, in policy gradient, the role of the value function is variance reduction. As SFT-PG also requires a value function during training, the computational overhead of DxMI is nearly identical to SFT-PG. Second, DxMI incorporates the maximum entropy principle, which facilitates exploration.

We also conduct ablation studies for the components of DxMI and append the results in Table 2 and 3. First, the temperature parameters are set to zero $\tau_1 = \tau_2 = 0$ in the sampler update (11). Then, we compare the linear time cost to the sigmoid time cost. In both cases, We observe the increase in FID and the decrease in Recall.

To investigate whether the trained value function captures useful information, we implement *value guidance*, where we shift the trajectory of generation slightly along the value function gradient, similarly to classifier guidance [47] and discriminator guidance [57]. When sampling the next step $\mathbf{x}_{t+1}$, we add a small drift with coefficient $\lambda$, i.e., $\mathbf{x}_{t+1} \leftarrow \mathbf{x}_{t+1} - \lambda \sigma_t \nabla_{\mathbf{x}_{t+1}} V_\psi(\mathbf{x}_{t+1})$. We observe sample quality metric improvement until $\lambda$ is 0.5. This observation suggests that the value function gradient is aligned well with the data density gradient.

## 5.3 Energy-Based Anomaly Detection and Localization

We demonstrate the ability of DxMI to train an accurate energy function on an anomaly detection task using the MVTec-AD dataset [61], which contains 224×224 RGB images of 15 object categories. We follow the multi-class problem setup proposed by [60]. The training dataset contains normal object images from 15 categories without any labels. The test set consists of both normal and defective object images, each provided with an anomaly label and a mask indicating the defect location. The goal is to detect and localize anomalies, with performance measured by AUC computed per object category. This setting is challenging because the energy function should reflect the multi-modal data distribution. Following the preprocessing protocol in [60, 59], each image is transformed into a 272×14×14 vector using a pre-trained EfficientNet-b4 [62]. DxMI is conducted in a 272-dimensional space, treating each spatial coordinate independently. With the trained energy function, we can evaluate the energy value of 14x14 spatial features and use max pooling and bilinear interpolation for anomaly detection and localization, respectively.

Table 5: MVTec-AD multi-class anomaly detection and localization experiment. Anomaly detection (DET) and localization (LOC) performance are measured in AUC. Due to the space constraint, only the average AUC over 15 classes is presented. The full results are provided in Table 6.

| Model | DET | LOC |
|---|---|---|
| DRAEM [58] | 88.1 | 87.2 |
| MPDR [59] | 96.0 | 96.7 |
| UniAD [60] | 96.5±0.08 | 96.8±0.02 |
| DxMI | **97.0** ±0.11 | **97.1**±0.02 |
| $\tau = 0$ | 67.9±5.90 | 84.6±4.02 |

We use separate networks for the energy function and the value function in this experiment, as the primary goal is to obtain an accurate energy function. We employ an autoencoder architecture for the energy function, treating the reconstruction error of a sample as its energy [63]. The diffusion model and the value function are five-step time-conditioned MLPs. Unlike conventional diffusion models, DxMI allows for a flexible choice of $\pi(\mathbf{x}_0)$. We set the initial distribution for the sampler to the data distribution applied with noise, aiming to identify the energy value more precisely near the data distribution. More experimental details can be found in Appendix C.3.

DxMI demonstrates strong anomaly classification and localization performance, as shown in Table 5. This result indicates that the trained energy function effectively captures the boundary of normal data. When entropy maximization is disabled by $\tau = 0$, the diffusion model fails to explore and only exploits regions of minimum energy, resulting in poor performance. We observe that a moderate level of $\tau = 0.1$ benefits both the sampler and the energy function, as it encourages exploration and provides a suitable level of diversity in negative samples.

# 6 Related Work

**Faster Diffusion Models.** Significant effort has been dedicated to reducing the number of generation steps in diffusion models during sampling while preserving sample quality. One popular approach is to keep the trained diffusion model unchanged and improve the sampling phase independently by tuning the noise schedule [8, 64, 65, 9], improving differential equation solvers [50, 66, 67, 68], and utilizing non-Markovian formulations [45, 69, 70]. While these methods are training-free, the sample quality can be further improved when the neural network is directly tuned for short-run sampling. Distillation methods train a faster diffusion sampler using training signal from a longer-run diffusion model, showing strong performance [10, 71, 72, 48, 73, 43, 42, 74]. A distilled model usually cannot outperform the teacher model, but adversarial or IRL methods may exceed full-length diffusion models. Hybrid methods [13, 12] combine distillation with adversarial loss, while other methods [46, 75] apply adversarial training to each denoising step. DxMI and SFT-PG [11] rely fully on adversarial training for final samples, allowing beneficial deviations from the diffusion path and reducing statistical distance from the data.

**RL for Diffusion Model.** RL is often employed to fine-tune diffusion models for a reward function. The source of the reward signal can be a computer program [24, 76, 20, 77, 78], or a human evaluator [19, 79, 25]. DxMI focuses on a setting where the estimated log data density is the reward. When RL is applied to diffusion models, the policy gradient [80] is the dominant choice [24, 76, 11, 77]. DxMI offers a value function-based approach as an alternative to the policy gradient. Maximum entropy RL for diffusion models is investigated in [20, 81, 82] but only in the continuous-time setting. DxDP investigates the discrete-time setting, which is more suitable for accelerating generation speed.

**Energy-Based Models.** DxMI provides a method of utilizing a diffusion model to eliminate the need for MCMC. Many existing EBM training algorithms rely on MCMC, which is computationally expensive and difficult to optimize for hyperparameters [34, 83, 18, 84, 85, 35, 63, 59]. Joint training of an EBM with a separate generative model is a widely employed strategy to avoid MCMC. EBMs can be trained jointly with a normalizing flow [86, 87], a generator [30, 88, 89], or a diffusion model [90, 33]. DxMI shares the objective function with several prior works in EBM [27, 28, 29, 30, 31, 33]. However, none of the works use a diffusion model directly as a sampler.

**Related Theoretical Analyses.** The convergence guarantees of entropy-regularized IRL are provided in [91, 92] under the assumption of a linear reward and the infinite time horizon. Their guarantees are not directly applicable to a practical instance of DxMI, mainly due to the nonlinearity of the reward function, the continuous state and action spaces, and the finite-horizon setting. Establishing the convergence guarantee for DxMI could be an important future research direction. On the other hand, theoretical analyses have been conducted on MaxEnt RL under finite state and action spaces [93], which is relevant for the discrete version of DxDP. More focused analysis on entropy regularized RL for diffusion models is provided in [94].

# 7 Conclusion

In this paper, we leverage techniques from sequential decision making to tackle challenges in generative modeling, revealing a significant connection between these two fields. We anticipate that this connection will spur a variety of algorithmic innovations and find numerous practical applications.

**Broader Impacts.** DxMI may facilitate deep fakes or fake news. However, trained on relatively low-resolution academic datasets, the models created during our experiments are not capable enough to cause realistic harm. Generative models trained solely using DxMI may possess fairness issues.

**Limitations.** Training multiple components simultaneously, DxMI introduces several hyperparameters. To reduce the overhead of practitioners, we provide a hyperparameter exploration guideline in Appendix B. DxMI is not directly applicable to training a single-step generator. However, a diffusion model fine-tuned by DxMI can be distilled to a single-step generator. DxMI does not offer the flexibility of using a different value of generation steps $T$ during the test time. Direct theoretical analysis of DxMI is challenging since the models are built on deep neural networks. Theoretical analysis that rationalizes the empirical results will be an important direction for future work.

## Acknowledgments and Disclosure of Funding

S. Yoon is supported by a KIAS Individual Grant (AP095701) via the Center for AI and Natural Sciences at Korea Institute for Advanced Study and IITP/MSIT (RS-2023-00220628). H. Hwang and F. Park are supported in part by IITP-MSIT grant RS-2021-II212068 (SNU AI Innovation Hub), IITP-MSIT grant 2022-220480, RS-2022-II220480 (Training and Inference Methods for Goal Oriented AI Agents), MSIT(Ministry of Science, ICT), Korea, under the Global Research Support Program in the Digital Field program(RS-2024-00436680) supervised by the IITP(Institute for Information & Communications Technology Planning & Evaluation), KIAT grant P0020536 (HRD Program for Industrial Innovation), SRRC NRF grant RS-2023-00208052, SNU-AIIS, SNU-IPAI, SNU-IAMD, SNU BK21+ Program in Mechanical Engineering, SNU Institute for Engineering Research, Microsoft Research Asia, and SNU Interdisciplinary Program in Artificial Intelligence. D. Kwon is partially supported by the National Research Foundation of Korea (NRF) grant funded by the Korea government (MSIT) (No. RS-2023-00252516 and No. RS-2024-00408003) and the POSCO Science Fellowship of POSCO TJ Park Foundation. Y.-K. Noh was partly supported by NRF/MSIT (No.RS-2024-00421203)) and IITP/MSIT (2020-0-01373).

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

# A   Adaptive Velocity Regularization

We are interested in the following optimization problem:

$$\min_{s_0,...,s_{T-1}} KL(\pi_\phi(\mathbf{x}_{0:T})||q_\theta(\mathbf{x}_T)\tilde{q}(\mathbf{x}_{0:T-1}|\mathbf{x}_T)). \tag{13}$$

Plugging in our choice of $\log \tilde{q}(\mathbf{x}_t|\mathbf{x}_{t+1}) = -\frac{||\mathbf{x}_t - \mathbf{x}_{t+1}||^2}{2s_t^2} - D\log s_t$, we can rewrite the optimization problem as follows.

$$\min_{s_0,...,s_{T-1}} \sum_{t=0}^{T-1} \mathbb{E}_{\mathbf{x}_t,\mathbf{x}_{t+1}\sim\pi}\left[\frac{||\mathbf{x}_t - \mathbf{x}_{t+1}||^2}{2s_t^2} + \frac{D}{2}\log s_t^2\right], \tag{14}$$

where constant term with respect to $s_t$ is omitted. Since the object function is separable, we can solve the optimization for each $s_t$ independently.

$$\min_{s_t} \frac{\mathbb{E}_{\mathbf{x}_t,\mathbf{x}_{t+1}\sim\pi}\left[||\mathbf{x}_t - \mathbf{x}_{t+1}||^2\right]}{2s_t^2} + D\log s_t, \quad t = 0,...,T-1. \tag{15}$$

This optimization has an analytic solution: $(s_t^*)^2 = \mathbb{E}_{\mathbf{x}_t,\mathbf{x}_{t+1}\sim\pi}\left[||\mathbf{x}_t - \mathbf{x}_{t+1}||^2\right]/D$.

# B   Guideline for Hyperparameters

**Value function.**   The most crucial design decision in DxMI is how to construct the value function. This design revolves around two primary axes. The first axis is whether the value function should be time-dependent or time-independent. The second axis is whether the value function should share model parameters with other networks, such as the sampler or the energy network.

Our experiments demonstrate various combinations of these design choices. In a 2D experiment, the time-dependent value function shares parameters with the EBM, which is a recommended approach for smaller problems. In an image experiment, we employ a time-independent value function that shares parameters with the energy network, effectively making the value function and the energy function identical. This design choice promotes monotonicity and efficient sample usage, as a single value function learns from all intermediate samples.

In an anomaly detection experiment, we use a time-dependent value function that does not share parameters with any other network. This design is suitable when a specific structure needs to be enforced on the energy function, such as with an autoencoder, and when making the structure time-dependent is not straightforward.

While these are the options we have explored, there are likely other viable possibilities for designing the value function.

**Coefficient $\tau$.**   Although the coefficient $\tau$ plays an important role in DxMI, we recommend running DxMI with $\tau = 0$ when implementing the algorithm for the first time. If everything is in order, DxMI should function to some extent. During normal training progression, the energy values of positive and negative samples should converge as iterations proceed.

After confirming that the training is progressing smoothly, you can start experimenting with increasing the value of $\tau$. Since $\gamma$ determines the absolute scale of the energy, the magnitude of $\tau$ should be adjusted accordingly. We set $\gamma = 1$ in all our experiments and recommend this value. In such a case, the optimal $\tau$ is likely to be less than 0.5.

**Learning rate.**   As done in two time-scale optimization [55], we use a larger learning rate for the value function than for the sampler. When training the noise parameters $\sigma_t$ in the diffusion model, we also assign a learning rate 100 times larger than that of the sampler.

# C   Details on Implementations and Additional Results

## C.1   2D Experiment

Sample quality is quantified using sliced Wasserstein-2 distance ($SW$) with 1,000 projections of 10k samples. The standard deviation is computed from 5 independent samplings. Density estimation

---
**Algorithm 2** Diffusion by Maximum Entropy IRL for Image Generation
---
1: **Input:** Dataset $\mathcal{D}$, Energy $E_\theta(\mathbf{x})$, Value $V_\psi(\mathbf{x}_t)$, and Sampler $\pi_\phi(\mathbf{x}_{0:T})$
2: $s_t \leftarrow \sigma_t$     // AVR initialization
3: **for** $\mathbf{x}$ in $\mathcal{D}$ **do**     // Minibatch dimension is omitted for brevity.
4:     Sample $\mathbf{x}_{0:T} \sim \pi_\phi(\mathbf{x})$.
5:     $\min_\theta E_\theta(\mathbf{x}) - E_\theta(\mathbf{x}_T) + \gamma(E_\theta(\mathbf{x})^2 + E_\theta(\mathbf{x}_T)^2)$     // Energy update
6:     **for** $t = T-1, \ldots, 0$ **do**     // Value update
7:         $\min_\psi [\text{sg}[V_\psi(\mathbf{x}_{t+1})] + R(t) - V_\psi(\mathbf{x}_t)]^2$
8:     **end for**
9:     **for** $\mathbf{x}_t$ randomly chosen among $\mathbf{x}_{0:T}$ **do**     // Sampler update
10:       Sample one-step: $\mathbf{x}_{t+1} \sim \pi_\phi(\mathbf{x}_{t+1}|\mathbf{x}_t)$     // Reparametrization trick
11:       $\min_\phi V_\psi^{t+1}(\mathbf{x}_{t+1}(\phi)) - \tau_1 D \log \sigma_t + \frac{\tau_2}{2s_t^2}||\mathbf{x}_t - \mathbf{x}_{t+1}(\phi)||^2$     // $\mathbf{x}_{t+1}$ is a function of $\phi$.
12:     **end for**
13:     $s_t^2 \leftarrow \alpha s_t^2 + (1-\alpha)||\mathbf{x}_t - \mathbf{x}_{t+1}||^2/D$     // AVR update
14: **end for**
---

performance is measured by AUC on discriminating test data and uniform samples over the domain. AUC is also computed with 10k samples.

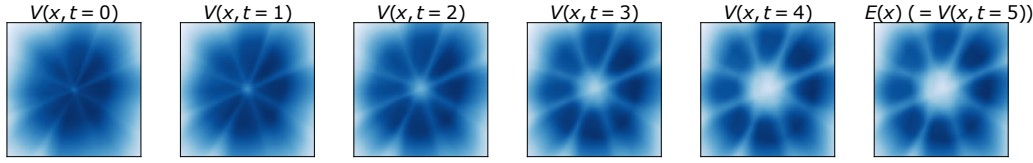

Figure 3: Value functions at each time step ($\tau = 0.1$ case). Blue indicates a low value.

## C.2 Image Generation

**Datasets.** CIFAR-10 is retrieved through torchvision API. ImageNet is downloaded from Kaggle and downsampled to $64 \times 64$ following `https://github.com/openai/guided-diffusion`. We apply random horizontal flips on all images. When computing FID, the whole 50,000 training images of CIFAR-10 are used. We make sure that no JPEG compression is used during processing. For ImageNet, we use the batch stat file provided by `https://github.com/openai/guided-diffusion`.

**Models.** For DDPM on CIFAR-10, we use the checkpoint provided by the SFT-PG repository [11]. For DDGAN, we utilize the official checkpoint. For EDM on ImageNet 64, we use the checkpoint from the consistency model repository [42]. In all experiments, we employ the same ResNet architecture, which does not include any normalization layers, such as batch normalization.

**Training.** For all runs, we use a batch size of 128. In the CIFAR-10 experiments, we use the Adam optimizer with a learning rate of $10^{-7}$ for the sampler weights, $10^{-5}$ for the value weights, and $10^{-5}$ for the $\sigma_t$'s. In the ImageNet 64 experiments, we use RAdam with a learning rate of $10^{-8}$ for the sampler. Additionally, we utilize a mixed precision trainer to handle FP16 weights. The value weights are updated with a learning rate of $10^{-5}$ using Adam. The $\sigma_t$'s are updated with a learning rate of $10^{-6}$. To select the best model, we periodically generate 10,000 images for CIFAR-10 and 5,000 images for ImageNet. The checkpoint with the best FID score is selected as the final model.

**Evaluation** For computing FID, Precision, and Recall scores, we used the TensorFlow-based evaluation script provided by Consistency Models [42] repository, which is based on the codebase of [47]. All the images are saved in a PNG format when fed to the evaluation script.

**Additional Details.** For optimal performance in image generation, we often tune the coefficients of two running costs separately. Let us denote the coefficient of $\log \pi(\mathbf{x}_{t+1}|\mathbf{x}_t)$ as $\tau_1$ and the coefficient of $\frac{1}{2s_t^2}||\mathbf{x}_t - \mathbf{x}_{t+1}||^2$ as $\tau_2$. In the CIFAR-10 experiments with $T = 10$ and $T = 4$, we set $\tau_1 = 0.1$

and $\tau_2 = 0.01$. In the ImageNet experiments with $T = 10$, we set $\tau_1 = \tau_2 = 0.01$, and for $T = 4$, we set $\tau_1 = 0.1$ and $\tau_2 = 0.01$.

We believe the optimal $\tau$ values vary for each setting due to differences in noise schedules and magnitudes. For example, the DDPM backbone is a variance-preserving formulation, while EDM is a variance-exploding formulation. Exploring a unified method for selecting the entropy regularization parameter is an interesting research direction.

### C.3 Anomaly Detection

**Dataset and Feature Extraction.** MVTec AD is a dataset designed to evaluate anomaly detection techniques, particularly for industrial inspection. It contains over 5000 high-resolution images divided into 15 different categories of objects and textures. Each category includes a set of defect-free training images and a test set with both defective and non-defective images.

The original dataset consists of high-resolution RGB images sized $224{\times}224$. Following the methods used in similar studies [59, 60], we extract a $272{\times}14{\times}14$ full feature representation of each image using a pre-trained EfficientNet-b4 [62]. We then train the energy function using DxMI in 272-dimensional space, treating every spatial feature from the training images as normal training data. Each data point $\mathbf{x}$ in $\mathbb{R}^{272}$ is projected to $\mathbb{S}^{272}$ through normalization. This projection is effective because the direction of the feature often represents the original image better than its magnitude.

**Anomaly Detection and Localization.** For anomaly detection, the energy of a single image is calculated by applying max pooling to the energy value of each spatial feature. For anomaly localization, the energy values of the $14{\times}14$ spatial features are upsampled to match the original $224{\times}224$ image resolution using bilinear interpolation.

**Model Design.** We utilize an autoencoder architecture for the energy function, as described in [63], using the reconstruction error of a sample as its energy value. Considering that the data distribution lies on $\mathbb{S}^{272}$, we appropriately narrow the function classes for the energy and sampler. Specifically, we constrain the decoder manifold of the energy function and the mean prediction of the sampler to remain on $\mathbb{S}^{272}$. We pretrain the energy function (i.e., the autoencoder) to minimize the reconstruction error of the training data. Both the sampler and the value function are trained from scratch.

**Choice of $\pi(\mathbf{x}_0)$.** Unlike traditional diffusion models, DxMI permits a flexible choice of $\pi(\mathbf{x}_0)$. To train the energy function effectively near the data distribution, we set the initial distribution for the sampler as the data distribution corrupted with noise. To apply meaningful noise to the data in $\mathbb{S}^{272}$, we use the pretrained autoencoder that is also used for the initial energy function to project the samples from the data distribution to the latent space, apply perturbations, and then restore the data to produce initial samples, as suggested in [59]. To maintain a consistent initial distribution, we fix the autoencoder used for generating the initial samples.

**Additional Details.** We use autoencoder with latent dimension 128 as the energy function. Encoder and decoder each consist of an MLP with 3 hidden layers and 1024 hidden dimensions. We use a time-conditional MLP for the sampler and value function, encoding the time information into a 128-dimensional vector using sinusoidal positional encoding. The input $\mathbf{x}_t$ is concatenated with the time embedding vector. We use MLPs with 3 hidden layers of 2048 and 1024 hidden dimensions for the sampler and value function, respectively.

The model is trained for 100 epochs with a batch size of 784 ($=4{\times}14{\times}14$) using the Adam optimizer. We use a learning rate of $10^{-5}$ for the sampler and value function and $10^{-4}$ for the energy function.

## D  Implementation and Computational Complexity of DxMI

DxMI (Algorithm 1) may seem complicated, but it largely mirrors the procedure of Max Ent IRL, with the exception of the AVR update. Notably, DxDP, the Max Ent RL subroutine within DxMI, is significantly simpler than standard actor-critic RL algorithms. Unlike these algorithms—such as Soft Actor Critic (SAC) [17], which requires training both a value function and two Q-functions—DxMI only trains a single value function and no Q-functions.

Table 6: MVTec-AD detection and localization task in the unified setting. AUROC scores (percent) are computed for each class. UniAD and DRAEM results are adopted from [60]. The largest value in a task is marked as boldface.

| | Detection | | | | Localization | | | |
| | DxMI | UniAD | MPDR | DRAEM | DxMI | UniAD | MPDR | DRAEM |
|---|---|---|---|---|---|---|---|---|
| Bottle | **100.0** ±0.00 | 99.7±0.04 | **100.0** | 97.5 | **98.5**±0.03 | 98.1±0.04 | **98.5** | 87.6 |
| Cable | **97.1**± 0.37 | 95.2±0.84 | 95.5 | 57.8 | 96.6±0.10 | **97.3**±0.10 | 95.6 | 71.3 |
| Capsule | 89.8± 0.61 | 86.9±0.73 | 86.4 | 65.3 | **98.5**±0.03 | **98.5**±0.01 | 98.2 | 50.5 |
| Hazelnut | **100.0**± 0.04 | 99.8±0.10 | 99.9 | 93.7 | **98.4**±0.04 | 98.1±0.10 | **98.4** | 96.9 |
| Metal Nut | 99.9± 0.11 | 99.2±0.09 | **99.9** | 72.8 | **95.5**±0.03 | 94.8±0.09 | 94.5 | 62.2 |
| Pill | **95.4**± 0.66 | 93.7±0.65 | 94.0 | 82.2 | **95.6**±0.07 | 95.0±0.16 | 94.9 | 94.4 |
| Screw | 88.9± 0.51 | 87.5±0.57 | 85.9 | **92.0** | **98.6**±0.08 | 98.3±0.08 | 98.1 | 95.5 |
| Toothbrush | 92.2±1.46 | **94.2**±0.20 | 89.6 | 90.6 | **98.8**±0.04 | 98.4±0.03 | 98.7 | 97.7 |
| Transistor | 99.2±0.28 | **99.8**±0.09 | 98.3 | 74.8 | 96.0±0.13 | **97.9**±0.19 | 95.4 | 65.5 |
| Zipper | 96.3±0.50 | 95.8±0.51 | 95.3 | **98.8** | 96.7±0.08 | 96.8±0.24 | 96.2 | **98.3** |
| Carpet | **99.9**±0.04 | 99.8±0.02 | **99.9** | 98.0 | **98.8**±0.02 | 98.5±0.01 | **98.8** | 98.6 |
| Grid | 98.6±0.28 | 98.2±0.26 | 97.9 | **99.3** | 97.0±0.09 | 96.5±0.04 | 96.9 | **98.7** |
| Leather | **100.0**±0.00 | **100.0**±0.00 | **100.0** | 98.7 | 98.5±0.03 | **98.8**±0.03 | 98.5 | 97.3 |
| Tile | **100.0**±0.00 | 99.3±0.14 | **100.0** | 95.2 | 95.2 ±0.14 | 91.8±0.10 | 94.6 | **98.0** |
| Wood | 98.3±0.33 | 98.6±0.08 | 97.9 | **99.8** | 93.8±0.07 | 93.2±0.08 | 93.8 | **96.0** |
| Mean | **97.0**±0.11 | 96.5±0.08 | 96.0 | 88.1 | **97.1**±0.02 | 96.8±0.02 | 96.7 | 87.2 |

DxMI does not demand excessive computation, a major concern in image generation experiments. In these experiments, the only additional component beyond the diffusion model is the EBM, which shares the same network as the value function. Additionally, the EBM used in DxMI is typically much smaller than the diffusion model, imposing minimal computational overhead. For instance, in our CIFAR-10 experiment (T=10), the EBM consists of 5M parameters, compared to 36M in the diffusion model. This is also significantly smaller than the critic networks used in GANs, such as the 158.3M parameters in BigGAN [49]. In practice, our CIFAR-10 experiment completes in under 24 hours on two A100 GPUs, while the ImageNet 64 experiment takes approximately 48 hours on four A100 GPUs.

## E Interpretation of the policy improvement method

In this section, we show that our policy improvement method introduced in Eq. (11) can effectively minimize the KL divergence between the joint distributions, $KL(\pi_\phi(\mathbf{x}_{0:T})||q_\theta(\mathbf{x}_T)\tilde{q}(\mathbf{x}_{0:T-1}|\mathbf{x}_T))$. The policy improvement at each timestep $t$ can be expressed as

$$\min_{\pi(\mathbf{x}_{t+1}|\mathbf{x}_t)} \mathbb{E}_{\mathbf{x}_t,\mathbf{x}_{t+1}\sim\pi} \left[ V^{t+1}(\mathbf{x}_{t+1}) + \tau \log \pi(\mathbf{x}_{t+1}|\mathbf{x}_t) - \tau \log \tilde{q}(\mathbf{x}_t|\mathbf{x}_{t+1}) \right] + const. \quad (16)$$

Here omitted the parameters $\phi$ and $\psi$ to avoid confusion. Using the definition of value function, above minimization can be expressed as

$$\min_{\pi(\mathbf{x}_{t+1}|\mathbf{x}_t)} \mathbb{E}_{\pi(\mathbf{x}_{t:T})} \left[ E(\mathbf{x}_T) + \tau \sum_{t'=t}^{T-1} \log \pi(\mathbf{x}_{t'+1}|\mathbf{x}_{t'}) - \tau \sum_{t'=t}^{T-1} \log \tilde{q}(\mathbf{x}_{t'}|\mathbf{x}_{t'+1}) \right], \quad (17)$$

where

$$\mathbb{E}_{\pi(\mathbf{x}_{t:T})} \left[ E(\mathbf{x}_T) + \tau \sum_{t'=t}^{T-1} \log \pi(\mathbf{x}_{t'+1}|\mathbf{x}_{t'}) - \tau \sum_{t'=t}^{T-1} \log \tilde{q}(\mathbf{x}_{t'}|\mathbf{x}_{t'+1}) \right] \quad (18)$$

$$= \tau\mathbb{E}_{\pi(\mathbf{x}_{t:T})} \left[ -\log q(\mathbf{x}_T) - \log \tilde{q}(\mathbf{x}_{t:T-1}|\mathbf{x}_T) + \log \pi(\mathbf{x}_{t:T}) - \log Z - \log \pi(\mathbf{x}_t) \right] \quad (19)$$

$$= \tau KL(\pi(\mathbf{x}_{t:T})||q(\mathbf{x}_T)\tilde{q}(\mathbf{x}_{t:T-1}|\mathbf{x}_T)) - \tau \log Z + \tau\mathcal{H}(\mathbf{x}_t) \quad (20)$$

Note that $\mathbf{x}_t$ is fixed in this optimization problem, and the optimization variable is $\mathbf{x}_{t+1}$. Therefore the policy improvement step at time $t$ is equivalent to

$$\min_{\pi(\mathbf{x}_{t+1}|\mathbf{x}_t)} KL(\pi(\mathbf{x}_{t:T})||q(\mathbf{x}_T)\tilde{q}(\mathbf{x}_{t:T-1}|\mathbf{x}_T)) \quad (21)$$

Therefore, the policy improvement step at each time step $t$ results in the minimization of the KL divergence between the joint distribution, $\pi(\mathbf{x}_{0:T})$ and $\tilde{q}(\mathbf{x}_{0:T})$.

# F  Additional Image Generation Results

Additional samples from the image generation models are presented in Fig. 4 and Fig. 5.

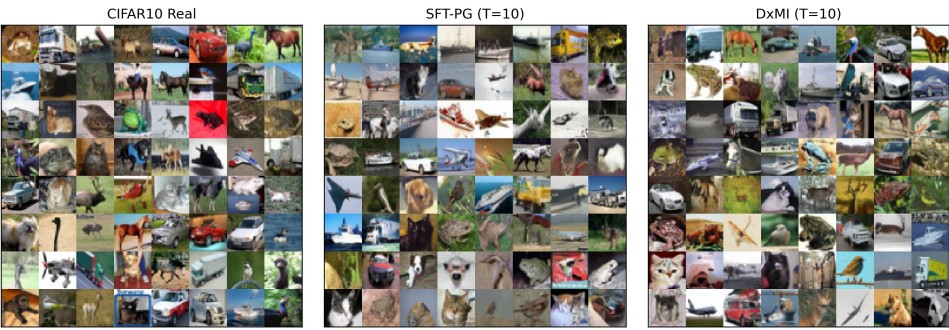

Figure 4: Randomly selected samples from CIFAR-10 training data, SFT-PG ($T = 10$, FID: 4.32), and DxMI ($T = 10$, FID: 3.19).

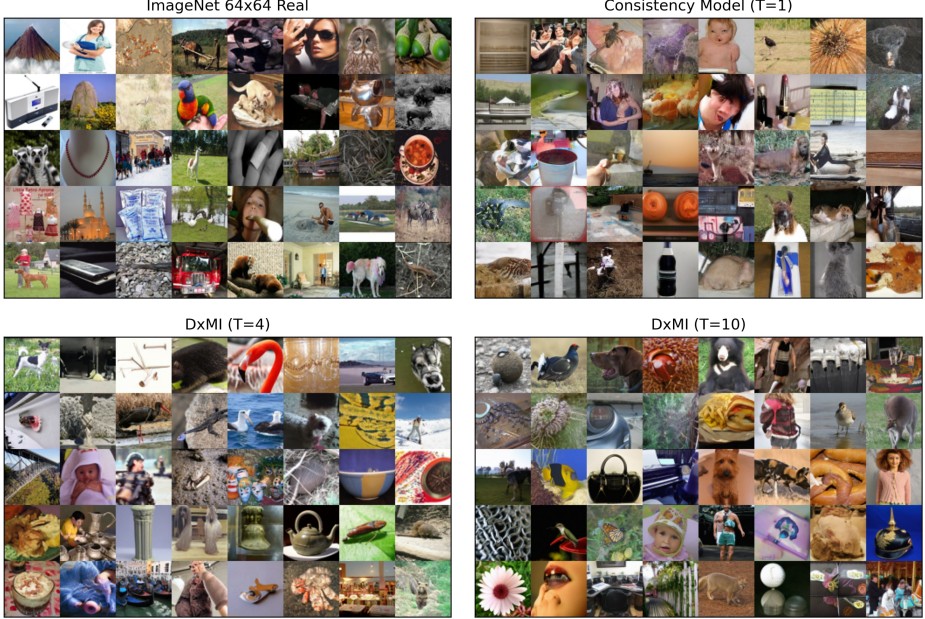

Figure 5: Randomly selected samples from ImageNet 64×64 training data, Consistency Model ($T = 1$, FID: 6.20), DxMI ($T = 4$, FID: 3.21), and DxMI ($T = 10$, FID: 2.68). Note that the Consistency Model samples distort human faces, while the DxMI samples depict them in correct proportions.

