# OpenReview forum: "Maximum Entropy Inverse Reinforcement Learning of Diffusion Models with Energy-Based Models"
_NeurIPS.cc/2024/Conference — NeurIPS 2024 oral_

### Official Review · Reviewer_YBX1 · 2024-07-08

**Soundness:** 3
**Presentation:** 3
**Contribution:** 3
**Rating:** 6
**Confidence:** 4

**Summary:**

This work proposes a maximum entropy inverse reinforcement learning (IRL) approach for improving the sample quality of diffusion generative models, especially when using a small number of generation steps.

**Strengths:**

1. The paper presents a novel and principled approach for improving diffusion models by formulating the problem as maximum entropy IRL. This elegant formulation allows jointly optimizing a diffusion model and EBM to enhance sample quality, especially with fewer generation steps.

2. The proposed DxDP algorithm is a key technical innovation that makes optimizing the IRL objective tractable. By leveraging ideas from optimal control and dynamic programming, DxDP enables efficient diffusion model updates without back-propagation through time, which is a significant practical benefit.

3. Strong empirical results on both image generation and anomaly detection tasks demonstrate the approach's effectiveness. DxMI can generate high-quality samples with very few (4-10) steps and enables training EBMs without MCMC, such a diffusion step reduction is impressive to me.

**Weaknesses:**

1. While the acclaimed acceleration of the diffusion model looks effective (and is also stated as a major advantage of this algorithm), the comparison to prior diffusion model acceleration methods is somewhat limited. A more comprehensive evaluation across different speed-quality tradeoffs and more discussion of DxMI's relative strengths and weaknesses compared to other approaches would be essential.

2. While the empirical results look good, the paper lacks theoretical analysis of the proposed methods, such as convergence rates for DxDP or any approximation guarantees relating the IRL objective to the original objective. Adding such analysis would help characterize this method's ups and downs.

**Questions:**

1. Upon reviewing this work, I found that the energy-based objective (Eq. (2)) and training objectives (Eqs. (3), (4), (5)) share very strong similarities/motivations to the concepts employed in existing KL-regularized RL-based fine-tuning papers (e.g., [1-4]). In particular, Eq. (6) and Eq. (7) in [2] serve precisely as the KL-based training objectives and the energy-based model, which involves sampling from unnormalized distributions. While I think the similarities might be just in terms of high-level methodologies and principles, it still becomes crucial for the authors to provide a thorough discussion highlighting the technical and methodological distinctions between this approach and the prior works in order to well-position the proposed method. For discussing the methodological relations/differences, I also recommend the authors carefully check [1] as this paper is written in a principled way, such that it becomes a good reference for understanding the methodologies.

2. As mentioned, do you have any theoretical insight into the convergence properties of DxDP or approximation guarantees relating to the IRL and original objectives? Many IRL theories might be worth checking (references [1-4] also might be worth checking for this goal, probably). Even without providing rates/bounds, an intuitive discussion/remark on these points could further validate the approach.

3. Can the DxMI approach be extended to other families of generative models, e.g., bridge-matching diffusion models, which are very similar to standard denoising diffusion models? What are the key challenges or requirements for the generative model?

4. How do you expect this approach to scale to more complex datasets or higher-dimensional spaces? Will the sample efficiency gains be more or less pronounced?

5. In experiments, how sensitive are the results to hyperparameters of DxMI/DxDP, such as the coefficient on the entropy term? What strategies did you use to tune these?

[1] https://arxiv.org/abs/2403.06279

[2] https://arxiv.org/abs/2402.16359

[3] https://arxiv.org/abs/2305.16381

[4] https://arxiv.org/abs/2405.19673

**Limitations:**

The authors adequately addressed the limitations.

---

> ### Author Rebuttal · Authors · 2024-08-06
>
> Dear Reviewer YBX1,
>
> We deeply appreciate your thorough and constructive comments. We will do our best to answer your questions.
>
> **Q. What is the connection to KL-regularized RL fine-tuning, particularly in papers such as [1-4]?**
>
> Thanks for pointing out an interesting connection. The sampler update step of DxMI is indeed closely related to KL-regularized RL.
>
> * We will augment our manuscript to discuss this connection in detail.
>     * We will add paragraphs dedicated to KL-regularized fine-tuning. In the paragraphs, we will cite the papers [1-4] and discuss them in detail.
>     * Please note that the current manuscript already mentions some works using KL-regularized RL, including [3], in the second paragraph of the Related Work section.
> * The sampler update objective of DxMI is a special case of KL-regularized RL. However, there are three key differences that makes DxMI distinct from existing KL-regularized RL fine-tuning methods.
>     * First, DxMI employs an uninformative reference policy, such as a Gaussian distribution (Eq. (7)). Due to this choice, DxMI can deviate from the pretraining model to find a better generation trajectory more suitable for small $T$.
>     * Second, DxMI employs a novel value-based algorithm for updating a diffusion model.
>     * Third, while most KL-regularized RL works assume that the reward function is known, DxMI simultaneously learns the reward from data.
> * BRAID [4] is particularly related to DxMI, as it also considers the problem of learning a reward from offline data. However, the reward is a separate random variable in [4], while in DxMI the reward is the log data density.
>
> [1] https://arxiv.org/abs/2403.06279 Tang. Fine-tuning of diffusion models via stochastic control: entropy regularization and beyond. 2024.
>
> [2] https://arxiv.org/abs/2402.16359 Uehara et al. Feedback Efficient Online Fine-Tuning of Diffusion Models. 2024.
>
> [3] https://arxiv.org/abs/2305.16381 Fan et al. DPOK: Reinforcement Learning for Fine-tuning Text-to-Image Diffusion Models. 2023.
>
>
> [4] https://arxiv.org/abs/2405.19673 Uehara et al. Bridging Model-Based Optimization and Generative Modeling via Conservative Fine-Tuning of Diffusion Models. 2024.
>
> **Q. The comparison to prior diffusion model acceleration methods is somewhat limited.**
>
> * Our comparative discussion had to be limited due to page constraints. Currently, the first paragraph of Section 6 ("Faster Diffusion Models") is dedicated to reviewing and comparing prior diffusion acceleration methods.
> * In the updated manuscript, we will enhance the "Faster Diffusion Models" section to highlight the key differences between DxMI and previous methods. Additionally, we will incorporate this comparison into the introduction and the experiments sections. The enhancements will include the following points:
>     * The key distinction between DxMI and existing diffusion acceleration methods is that DxMI does not use the intermediate states in the trajectory of a diffusion model. Most diffusion distillation methods focus on preserving or learning from these intermediate states. In contrast, DxMI directly aims to match the final state of a sampler to the data distribution. The promising performance of DxMI indicates that deviating from the original diffusion trajectory may be beneficial for sample quality when the generation has to be performed in a very few steps. Among existing methods, only SFT-PG employs a similar approach; however, DxMI outperforms SFT-PG by using dynamic programming instead of a policy gradient.
>
>
> **Q. A more comprehensive evaluation across different speed-quality tradeoffs would be essential.**
>
> * In the updated manuscript, we will include a figure demonstrating the speed-quality trade-off with more data points, such as $T$=2, 20, and 40.
> * Qualitatively, the best trade-off for DxMI is achieved in the mid-range of $T$, from 4 to 10.
>     * If $T$ is too small, the sampler is less capable, and the data processing inequality becomes less tight, making our MaxEnt regularization less effective.
>     * If $T$ is too large, the sampler's capability increases, but the value function learning becomes more challenging.
>
> **Q. More discussion of DxMI's relative strengths and weaknesses compared to other approaches would be essential.**
>
> We will augment our discussion on strengths and weaknesses of DxMI in the update manuscript. Currently, some of weaknesses are mentioned in our Limitation section. Focusing on diffusion acceleration application, our strengths and weaknesses can be summarized as follows.
> * Strengths
>     * Unlike other diffusion distillation methods where the performance is bounded by the teacher model, in principle DxMI may achieve better sample quality than the pretrained model (see our CIFAR-10 case).
>     * The dynamic programming-based in DxMI is more effective than the policy gradient-based algorithm (e.g., SFT-PG).
>     * DxMI produces an EBM as a byproduct, which can be used in other applications such as anomaly detection or transfer learning.
> * Weaknesses
>     * When $T=1$, DxMI reduces to GAN, offering no additional advantage.
>     * DxMI does not offer the flexibility of using a different value of $T$ during the test time.
>
> ----
> Due to the character limit, we will continue answering your question in the comment.

---

> ### Author Response · Authors · 2024-08-06
> **Continued Response for Reviewer YBX1**
>
> **Q. Theoretical analysis of DxMI**
>
> * We agree that theoretical analysis of our MaxEnt IRL problem would be very interesting. However, direct analysis of DxMI may be highly challenging because our EBM and diffusion model comprise deep neural networks.
> * Conducting theoretical analysis in a more restricted setting might be more feasible while still providing valuable insights.
> * To the best of our knowledge, there is not many theoretical results for MaxEnt IRL. Previous works [5,6] offered convergence guarantees for MaxEnt IRL in a discrete state-action space. While their results do not directly apply to DxMI due to algorithmic and other differences, their results suggest that similar analysis could be conducted on DxMI under suitable assumptions.
> * Please consider that the main focus of this paper is to present a practical algorithm that is empirically scalable and effective. We believe providing theoretical analysis that rationalizes the empirical results in the paper is an important future work.
> * We will add a paragraph describing the theoretical results from [5,6] in our Related Work section. We will also mention the difficulty of theoretical analysis in our limitations section.
>
>
> [5] Renard et al., Convergence of a model-free entropy-regularized inverse reinforcement learning algorithm, arxiv, 2024.
>
> [6] Zeng et al., Maximum-Likelihood Inverse Reinforcement Learning with Finite-Time Guarantees, NeurIPS 2022.
>
> **Q. Can the DxMI approach be extended to other families of generative models, e.g., bridge-matching diffusion models?**
>
> * Thanks for the interesting suggestion. We do believe the value-based learning presented in DxMI can be extended to other types of generative modeling problems, such as finding a bridge between distributions.
>
> **Q. How do you expect this approach to scale to more complex datasets or higher-dimensional spaces? Will the sample efficiency gains be more or less pronounced?**
>
> * Currently, we do not see a particular obstacle that prevents DxMI from scaling to larger datasets. We believe DxMI is at least as scalable as GANs, which have already demonstrated its feasiblity in a very high-dimensional datasets. As an empirical evidence, we will add LSUN Bedroom 256x256 experiments in the updated manuscript.
>
> **Q. How sensitive are the results to hyperparameters of DxMI/DxDP, such as the coefficient on the entropy term? What strategies did you use to tune these?**
>
> * DxMI is not sensitive to the entropy regularization coefficient $\tau$, which can be safely set to 0.01 or 0.1. This insensitivity arises because the energy, which competes with the entropy, is regularized by the coefficient $\gamma$ to maintain a narrow range of values close to zero.
> * We provide the guide for hyperparameter tuning in Appendix B.
> * Probably the most critical hyperparameter is learning rates, for which we assign a larger value for the energy and a smaller value for the diffusion model. This is included in Appendix B.
>
>
>
> Thank you once more for your insightful feedback. We hope our responses have addressed your concerns. Please feel free to reach out if you have any further questions or need additional information.
>
> Best regards,
> Authors.

---

> > ### Comment · Reviewer_YBX1 · 2024-08-09
> >
> > Thanks for the detailed and informative response. The strengths of this work and connections with relevant topics are made much clearer.

---

> > > ### Author Response · Authors · 2024-08-09
> > > **Thanks for your reply**
> > >
> > > Thanks for acknowledging the strength of this work. Also, thanks again for bringing up the interesting connection to existing work. We will make sure the connection is described well in the updated version.
> > >
> > > Best regards,
> > >
> > > Authors.

---

### Official Review · Reviewer_yjrf · 2024-07-11

**Soundness:** 3
**Presentation:** 3
**Contribution:** 3
**Rating:** 7
**Confidence:** 3

**Summary:**

The authors propose learning a denoising diffusion model without using the denoising loss. Instead, they propose first training an energy based model and then treating the diffusion denoising sampler as an RL trajectory with the energy based model as the reward. To learn the energy based model, however they propose a generalized version of contrastive divergence which uses the current diffusion model as part of its objective function. Experiments are designed to validate this idea, starting from pretrained diffusion checkpoints.

**Strengths:**

- The paper explores a creative combination of a variety of ideas all seeking to replace the simple denoising diffusion training objective.
- The exposition is well written
- The proposed approach involves learning both an energy based model and a diffusion model using DP where the number of steps can be small. The latter has value for fine-tuning diffusion models against other types of rewards.

**Weaknesses:**

- The authors introduce an elaborate scheme to do away with the score function denoising objective, but their experiments rely on pre-trained checkpoints that use the score function training to get these results so the final results are based on stacking the new methods on top of the original diffusion training.

- The objective in Eq (5) seems somewhat hard to be confident of, in the sense that given any $\pi$ and $p$, the optimal $q$ could diverge away from $p$ such that $KL(p||q) < KL(\pi|q) >> 0$ (while still being closer than q since the optimum value of the objective has to be negative). This could theoretically make it unstable as an iteration progresses, $\pi$ will track $q$ and then $q$ can take another step further away from both $p$ and $\pi$ while being optimal for Eq (5), which only requires that the relative distance to $p$ be less than that of $q$.  As another point, when initializing with a $\pi$ that is close to $p$ (e.g. in the pretrained model case), the optimal $q$ does not look like it has to be close to either given the cancellation involved.

-  The key proposal in Algorithm 1 has a multiple highly unstable procedures mixed together within each iteration -- e.g. the energy update in Line 5 and the TD bootstrap objective in line 7.

**Questions:**

In Algorithm 1, Line 5, are the $x$ and $x_T$ completely independent samples with no relation to each other? From the way its written, $x_T$ is sampled starting from independent noise $x_0$, and $x$ is a particular data sample for that iteration.  Moreover, since for the pre-trained checkpoints, $\pi$ is likely close to $p$, the energy objective looks like it has minimal training signal.

**Limitations:**

Yes

---

> ### Author Rebuttal · Authors · 2024-08-06
>
> Dear Reviewer yjrf,
>
> We appreicate your time and effort in reviewing our work. Here, we are happy to address your concerns and questions.
>
> **Q. Is the objective function stable?**
>
> > The objective in Eq (5) seems somewhat hard to be confident of, in the sense that given any $\pi$ and $p$, the optimal $q$ could diverge away from p such that $KL(p||q) = KL(\pi||q) \gg 0$.  This could make it unstable as an iteration progresses, $\pi$ will track q and then q can take another step further away from both p and $\pi$ while being optimal for Eq (5).
>
> We would like to clarify the critical misunderstandings regarding the objective function.
>
> Let us write our objective function as $L(q,\pi)=KL(p \parallel q) - KL(\pi \parallel q)$, where we aim to solve $\min_q \max_\pi L(q,\pi)$.
>
> * When $p$ and $\pi$ are fixed and $p\neq \pi$, the optimal $q$ does not satify $KL(p \parallel q) = KL(\pi \parallel q) \gg 0$.
>     * It is a misunderstanding that our objective function $L(q,\pi)$ has a minimum at 0, where $KL(p||q) = KL(\pi||q)$.
>     * When $p\neq \pi$, the objective function can have a negative value. For example, setting $q=p$ makes the objective negative $L(p,\pi)=-KL(p \parallel \pi)<0$. Therefore, $KL(p||q) = KL(\pi||q)$ is not optimal for $q$, as there are other values of $q$ that achieve a lower objective function value.
>     * Thus, our objective function always keep $q$ closer to $p$ than to $\pi$, not invoking the instability questioned in the review comment.
>
>
> **Q. Is learning possible when $\pi$ is close to $p$?**
> > As another point, when initializing with a $\pi$ that is close to p (e.g. in the pretrained model case), the optimal q does not look like it has to be close to either given the cancellation involved.
>
> > Moreover, since for the pre-trained checkpoints, π is likely close to p, the energy objective looks like it has minimal training signal.
>
> * When $p=\pi\neq q$, it is true that there is no learning signal for $q$. However, $p=\pi\neq q$ is not a Nash equilibrium, and learning is not terminated at this point.
> * When $p=\pi\neq q$, our objective function drives $\pi$ away from $p$ to be close to $q$. After $\pi$ becomes different from $p$, the learning signal for $q$ is generated.
>     * This behavior may be undesirable in practice. However, we are most interested in the case where the number of function evaluations is small ($T=4$ or $10$). With small $T$, the initial sample quality from $\pi$ is very bad (Figure 1 (Right) of the manuscript), indicating that $p \neq \pi$.
> * As $p=\pi\neq q$ is not a Nash equilibrium of our objective function, an optimization algorithm, if done correctly, will eventually lead to the Nash equilibirum $p=q=\pi$.
>
>
> **Q. DxMI still uses a diffusion model as a starting point.**
> > The authors introduce an elaborate scheme to do away with the score function denoising objective, but their experiments rely on pre-trained checkpoints that use the score function training to get these results so the final results are based on stacking the new methods on top of the original diffusion training.
>
> * DxMI is not meant to be a complete replacement for denoising objective. We will update our introduction to clarify that DxMI is a complementary training algorithm for diffusion models.
> * Please note that diffusion model pre-training is not always necessary for DxMI. In our 2D experiment and anomaly detection experiment, DxMI demonstrated its ability to train a sampler without pre-training.
>
>
> **Q. The proposed algorithm has multiple unstable procedures.**
>
> > The key proposal in Algorithm 1 has a multiple highly unstable procedures mixed together within each iteration -- e.g. the energy update in Line 5 and the TD bootstrap objective in line 7.
>
> * We understand the proposed algorithm can be seemingly complicated.
>     * To deal with the complexity, we will make our code public and provide model checkpoints. Also, we disclose our hyperparameters and suggest a hyperparameter selection strategy in Appendix B.
> * However, to the best of our knowledge, there is no empirical or theoretical evidence of instability.
>     * The TD update and the energy update does not interfere with each other, as they operate on different inputs. Both updates are stable, as the energy update equation is regularized (coefficient $\gamma$), and TD update is simply mean squared error minimization.
> * Empirically, we found the algorithm much more stable than MCMC-based EBM training algorithms, which occasionally diverge for no reason.
> * If you have a particular concern regarding the algorithm's stability, we are happy to discuss it.
>
>
> **Q. In Algorithm 1, Line 5, are the x and x_T completely independent samples with no relation to each other?**
>
> * You are correct that, in Algorithm 1, $\mathbf{x}$ denotes a real data and $\mathbf{x}_T$ indicates a sample from the diffusion model. We will make this point clear in the updated manuscript by adding a comment in Algorithm 1.
>
>
>
>
> Thank you again for your constructive review. We believe there was a misunderstanding regarding our objective function, and we hope our response clarifies this issue. We kindly request that you reconsider the decision in light of our responses and update the score accordingly. We are also eager to address any additional concerns you may have.
>
> Best regards,
> Authors.

---

> > ### Comment · Reviewer_yjrf · 2024-08-08
> > **Reply**
> >
> > > Thus, our objective function always keep closer to than to, not invoking the instability questioned in the review comment.
> >
> > Thanks for the clarification, I understand the proposal better after re-reading the paper and have updated the initial review accordingly.
> >
> > While my comment related to the objective was not really trying to claim a formal counter example for demonstrating pathologies of the proposed objective, it would be quite impressive if you could convert your intuition in the above response into a formal argument backing the proposal.

---

> ### Author Response · Authors · 2024-08-09
> **Thank you for your reply.**
>
> Thank you very much for taking the time to revisit the manuscript. We truly appreciate your reconsideration of the score. We will make sure your comments are reflected in the manuscript well and try to come up with some formal statements that we can make to ensure the stability of the objective function.
>
> Best regards,
> Authors.

---

### Official Review · Reviewer_Nqik · 2024-07-12

**Soundness:** 3
**Presentation:** 4
**Contribution:** 4
**Rating:** 8
**Confidence:** 3

**Summary:**

This paper seeks to improve diffusion models by employing inverse reinforcement learning methods of imitation rather than (more myopic) behavioral cloning methods, which prevalent existing diffusion models can be viewed as using. It trains an energy-based model using maximum entropy inverse reinforcement learning and proposes an optimal control problem for diffusion based on minimizing an upper bound of the contrastive KL divergence. The benefits of the approach are demonstrated with a focus on generating outputs with few diffusion iterations.

**Strengths:**

The paper is motivated by the key connection between imitation and diffusion models, the characterization of existing methods corresponding to behavioral cloning approaches for imitations, and the potential for improved diffusion using more sophisticated inverse reinforcement learning imitation methods. This motivation is nicely described.

The technical content of the paper is quite dense, but the authors present it clearly.

Other work exists exploring this perspective of diffusion as imitation/optimal control, but the paper’s approach is nicely constructed to avoid MCMC and policy gradient optimization, which are often bottlenecks in existing methods.

Experimental results show the benefits of this approach, including comparisons with a similar approach that is reliant on policy gradient (SFT-PG) and other recent diffusion model learning methods for generation and anomaly detection.

**Weaknesses:**

Closely related work isn’t described in the introduction to better frame the contributions of this paper.

**Questions:**

Is there potential for theoretical analysis or guarantees using this approach?

Are there visual differences in the generated outputs of different models that could be highlighted in 5.2?

**Limitations:**

Potential abuses using deep fakes are described, along with limitations.

---

> ### Author Rebuttal · Authors · 2024-08-06
>
> Dear Reviewer Nqik,
>
> Thank you for taking the time to review our work. We deeply value your feedback and are happy to address your questions.
>
> **Q. Is theoretical analysis possible?**
> > Is there potential for theoretical analysis or guarantees using this approach?
>
> Thanks for bringing up an important point.
>
> * Yes, there is significant potential for theoretical analysis in our MaxEnt IRL problem. However, directly analyzing the DxMI implementation may be highly challenging because our EBM and diffusion model comprise deep neural networks.
> * Conducting theoretical analysis in a more restricted setting might be more feasible while still providing valuable insights. Previous works [1,2] offered convergence guarantees for MaxEnt IRL in a discrete state-action space. While their results do not directly apply to DxMI due to algorithmic and other differences, their results suggest that a similar analysis could be conducted on DxMI under suitable assumptions.
> * Please consider that the main focus of this paper is to present a practical algorithm that is empirically scalable and effective. We believe providing a theoretical analysis that rationalizes the empirical results in the paper is an important future work.
> * We will add a paragraph describing the theoretical results from [1,2] in our Related Work section. We will also mention the difficulty of theoretical analysis in our limitations section.
>
>
> [1] Renard et al., Convergence of a model-free entropy-regularized inverse reinforcement learning algorithm, arxiv, 2024.
>
> [2] Zeng et al., Maximum-Likelihood Inverse Reinforcement Learning with Finite-Time Guarantees, NeurIPS 2022.
>
> **Q. Related works are not described in the introduction.**
>
> > Closely related work isn’t described in the introduction to better frame the contributions of this paper.
>
>
> * We had to defer our discussion on prior works to the Related Work section (Section 6) due to the page limitation.
> * In the revised manuscript, we will include a paragraph in the introduction which describes related work. If there is any specific work that you want us to additionally cite, we are happy to incorporate them into the updated manuscript.
>
> **Q. Are there visual differences in the generated outputs of different models?**
>
> * Please find examples of randomly generated images from DxMI and other models in the attached PDF, highlighting the visual differences in their outputs. For example, the Consistency Model samples distort human faces, while the DxMI samples depict them in correct proportions.
>
> Again, we thank the reviewer for acknowledging the value of our work. If you have any further questions, we are happy to address them.
>
> Best regards,
> Authors.

---

> > ### Comment · Reviewer_Nqik · 2024-08-14
> >
> > Thank you for addressing my concerns.

---

### Official Review · Reviewer_EhWr · 2024-07-12

**Soundness:** 3
**Presentation:** 3
**Contribution:** 3
**Rating:** 6
**Confidence:** 3

**Summary:**

The authors introduce a maximum entropy inverse reinforcement learning (IRL) approach for
enhancing the sample quality of diffusion generative models, especially with limited generation
time steps. Named Diffusion by Maximum Entropy IRL (DxMI), the approach involves joint
training of a diffusion model and an energy-based model (EBM). The EBM provides the
estimated log density as a reward signal for the diffusion model, which is trained to maximize
both the reward from EBM and the entropy of generated samples. Additionally, the authors
propose Diffusion by Dynamic Programming (DxDP), a novel reinforcement learning algorithm
that optimizes diffusion model updates efficiently by transforming the problem into an optimal
control formulation. Empirical studies demonstrate that diffusion models fine-tuned with DxMI
can generate high-quality samples in as few as 4 to 10 steps and improve the stability of EBM
training dynamics, enhancing anomaly detection performance.

**Strengths:**

1. Innovation: The proposed DxMI methodology introduces the concept of maximum entropy IRL to the training of diffusion models, which is novel and potentially impactful in improving sample quality and inference time of diffusion model.
2. Clarity: The manuscript is well-written and logically structured, with clear explanations of the theoretical foundations and algorithms.

**Weaknesses:**

1. Complexity of Implementation: The implementation of DxMI, particularly the joint training
of diffusion models and EBMs, might be complex and require significant computational
resources. The practical feasibility in various settings could be further elaborated.
2. Limited Scope of Experiments: The experiments, while promising, are somewhat limited in
scope. More diverse and complex tasks could further validate the robustness and versatility
of the proposed approach.
3. Comparative Analysis: While the proposed methods show improvements, a more detailed
comparative analysis with state-of-the-art techniques, including training time and
computational cost comparisons, would strengthen the paper.

**Questions:**

1. Generalization to Complex Tasks: How do the authors envision the performance of DxMI in
more complex generative tasks, such as high-resolution image generation or text-to-image
synthesis?
2. Training Time Comparison: What is the average training time for models using DxMI
compared to other generative model training techniques such as GANs or VAE-based
approaches?
3. Scalability: How scalable is the proposed DxMI approach when applied to larger datasets or
models with significantly more parameters? Are there any anticipated bottlenecks?

**Limitations:**

The authors have clearly presented the limitations in the paper.

---

> ### Author Rebuttal · Authors · 2024-08-06
>
> Dear Reviewer EhWr,
>
> We appreciate the comprehensive feedback on our manuscript. All the comments and questions raised have been considered below.
>
>
> **Q. Complexity of Implementation**
> > The implementation of DxMI, particularly the joint training of diffusion models and EBMs, might be complex and require significant computational resources.
>
> * DxMI may seem complicated, but in fact its complexity is no larger than GANs and actor-critic RL methods.
>     * Particularly in image experiments, EBM is the only additional component over a diffusion model. EBM functions similarly to the discriminator of GAN. Furthermore, EBM used in DxMI is typically much smaller than the diffusion model, introducing minimal burden. For example, in our CIFAR-10 experiment (T=10), the EBM has 5M parameters while the diffusion model has 36M parameters.
>     * DxMI is significantly simpler than standard actor-critic RL, such as Soft Actor-Critic (SAC) [1], which trains a value function and two Q functions simultaneously. On the contrary, DxMI does not require a Q function and trains only a single value function.
>
> [1] Haarnoja, Tuomas, et al. "Soft actor-critic: Off-policy maximum entropy deep reinforcement learning with a stochastic actor." ICML 2018. https://proceedings.mlr.press/v80/haarnoja18b
>
>
> **Q. More Experiments**
>
> >The practical feasibility in various settings could be further elaborated.
>
>
> > The experiments, while promising, are somewhat limited in scope. More diverse and complex tasks could further validate the robustness and versatility of the proposed approach.
>
> > How do the authors envision the performance of DxMI in more complex generative tasks, such as high-resolution image generation or text-to-image synthesis?
>
> * The manuscript already includes experiments on a variety of tasks and data types, such as 2D density estimation, unconditional and conditional image generation, and anomaly detection using latent vectors. If you find any particular aspect of the experimental portfolio to be limited, please let us know.
> * To further demonstrate the scalability of DxMI, we will provide additional experimental results on LSUN Bedroom 256x256. DxMI (T=4) achieves competitive results (FID: 5.93, Recall: 0.477), whereas DDPM (T=1000) achieves FID: 4.89, Recall 0.45. We will augment our experiment section with the LSUN Bedroom experiment. This observation shows that DxMI is indeed scalable to high-dimensional data.
> * We believe that DxMI can be effectively applied to text-to-image synthesis. However, text-to-image synthesis typically requires a significant amount of computing resources (at least 25 A100 days) and involves numerous experimental conditions to explore (e.g., selecting text prompts). Therefore, we suggest this as an intriguing direction for future research.
>
>
> **Q. Comparative Analysis**
> > While the proposed methods show improvements, a more detailed comparative analysis with state-of-the-art techniques.
>
> * Please find that the current manuscript provides comparative analysis on Section 6 Related Work.
> * We will augment our comparative analysis by adding additional paragraphs in the introduction and sections that describe the proposed method.
>
> **Q. Average Training Time**
>
> > What is the average training time for models using DxMI compared to other generative model training techniques such as GANs or VAE-based approaches?
>
> * Our CIFAR-10 experiment takes less than 24 hours on two A100 GPUs, while our ImageNet 64 experiment takes approximately 48 hours on four A100 GPUs. The computational resources required for DxMI training are significantly lower compared to state-of-the-art GANs, which can take up to 48 hours on a Google TPU V3 Pod [2]. A TPU Pod consists of 1024 TPU chips, which are considerably more powerful than a few A100 GPUs. This difference is partly because DxMI can leverage a pre-trained diffusion model.
>
> [2] Brock et al. Large scale GAN training for high fidelity natural image synthesis. ICLR 2019. https://arxiv.org/abs/1809.11096
>
> **Q. Scalability**
> > Scalability: How scalable is the proposed DxMI approach when applied to larger datasets or models with significantly more parameters? Are there any anticipated bottlenecks?
>
> * For now, we do not observe any sign of bottleneck or inscalability. We believe DxMI is as scalable as GANs, since DxMI also leverages an EBM as a discriminator. GANs have already been shown to be scalable to very high-dimensional data [2].
>
>
> Again, we thank you for providing valuable comments. Do not hesitate to let us know if you have further questions.
>
> Best wishes,
> Authors.

---

> > ### Comment · Reviewer_EhWr · 2024-08-09
> >
> > Thanks for the detailed responses. All of my concerns have been addressed properly.

---

### Author Rebuttal · Authors · 2024-08-06

## General Comments to AC and All Reviewers

We appreciate all reviewers for their thoughtful comments and remarks. We thank the reviewers for their insightful feedback and constructive comments and for providing suggestions that would improve our paper.

First of all, we are encouraged that the reviewers found the following points:

(i) Our approach and formulation are novel, principled, and elegant in improving sample quality and inference time of a diffusion model. (EhWr, YBX1)

(ii) The motivation is nicely described, the algorithm is well constructed to avoid bottlenecks in existing methods (Nqik), and it uses a creative combination of ideas (yjrf).

(iv) The experimental results show the benefits of the proposed approach (Nqik), which are strong and impressive (YBX1).

(iii) The manuscript is well-structured, and the presentation is clear. (EhWr, Nqik, yjrf)


Based on the feedback from all reviewers, the most significant shared concerns or major points to address are as follows.

**On theoretical analysis (Nqik, YBX1):** We agree with the reviewers that a theoretical analysis of our MaxEnt IRL problem would be valuable. To the best of our knowledge, there are few theoretical results available for MaxEnt IRL. Previous works [1,2] have provided convergence guarantees for MaxEnt IRL in a discrete state-action space with an infinite horizon. In contrast, our approach considers a continuous state-action space with a finite horizon. Additionally, there are algorithmic differences regarding the reward functions. Consequently, their results do not directly apply to DxMI. However, we believe a similar analysis could be performed on DxMI under appropriate assumptions and a simplified setting, particularly with linear reward functions and a tabular policy.

[1] Renard et al., Convergence of a model-free entropy-regularized inverse reinforcement learning algorithm, arxiv, 2024.

[2] Zeng et al., Maximum-Likelihood Inverse Reinforcement Learning with Finite-Time Guarantees, NeurIPS 2022.


**On scalability and complexity of the algorithm (EhWr, YBX1):** Because we formulate the problem using MaxEnt IRL, DxMI might initially seem complex. However, its complexity is comparable to Generative Adversarial Networks (GANs) and actor-critic reinforcement learning methods. We believe that DxMI is as scalable as conventional diffusion models and GANs, which have already demonstrated scalability to very high-dimensional data. To demonstrate the scalability of DxMI, we run DxMI (T=4) on LSUN Bedroom 256x256 and achieve competitive results (FID: 5.93, Recall: 0.477), where DDPM (T=1000) achieves FID: 4.89, Recall 0.45. Additionally, we have not observed any signs of bottlenecks or scalability issues. For instance, DxMI is not particularly sensitive to the entropy regularization coefficient. We provide guidance for hyperparameter tuning in Appendix B. More specific answers can be found in individual comments.


The attached PDF provides examples of generation from DxMI on CIFAR-10 and ImageNet 64x64, along with samples from competitive baselines.

We hope the above answers some common questions from the reviewers. We also respond to individual comments from each reviewer below.

---

### Decision · Program_Chairs · 2024-09-25

**Decision:**

Accept (oral)

**Comment:**

The paper presents a novel and efficient methodology to improve diffusion generative models based on the formulation of an maximum entropy inverse RL problem.

All reviewers agree this paper presents a novel and significant contribution which is methodologically strong.

During the author's rebuttal several points were discussed in detail.
In order of importance, the points revolved around:

(1) The complexity and scalability of the algorithm.

(2) The existence of theoretical guarantees / formal statements to ensure the stability of the objective function.

(3) Additional comparative analysis to prior diffusion model acceleration methods with relative strengths and weaknesses.

(4) The need for additional experiments.

(5) A more comprehensive evaluation across different speed-quality trade-offs.

(6) Misunderstandings related to the objective function, the need of a pre-trained diffusion model.

The rebuttal was convincing in clarifying all these points and no further discussion was required.

I think this is a very strong paper which will have significant impact and thus recommend acceptance.

While the primary focus of this work is to introduce a practical algorithm, I suggest incorporating the discussion regarding the above points as much as possible.